# Strongly enhanced THz generation enabled by a graphene hot-carrier fast lane

Dehui Zhang [1], Zhen Xu [1], Gong Cheng [1], Zhe Liu[1], Audrey Rose Gutierrez [1], Wenzhe Zang[1], Theodore B. Norris [1] ✉ & Zhaohui Zhong [1] ✉

Semiconductor photoconductive switches are useful and versatile emitters of terahertz (THz) radiation with a broad range of applications in THz imaging and time-domain spectroscopy. One fundamental challenge for achieving efficient ultrafast switching, however, is the relatively long carrier lifetime in most common semiconductors. To obtain picosecond ultrafast pulses, especially when coupled with waveguides/transmission lines, semiconductors are typically engineered with high defect density to reduce the carrier lifetimes, which in turn lowers the overall power output of the photoconductive switches. To overcome this fundamental trade-off, here we present a new hybrid photoconductive switch design by engineering a hot-carrier fast lane using graphene on silicon. While photoexcited carriers are generated in the silicon layer, similar to a conventional switch, the hot carriers are transferred to the graphene layer for efficient collection at the contacts. As a result, the graphene-silicon hybrid photoconductive switch emits THz fields with up to 80 times amplitude enhancement compared to its graphene-free counterpart. These results both further the understanding of ultrafast hot carrier transport in such hybrid systems and lay the groundwork toward intrinsically more powerful THz devices based on 2D-3D hybrid heterostructures.

The photoconductive switch (PCS) is an important and widely used technology in pulsed THz field generation and detection, making it a critical building block in THz imaging and time-domain spectroscopy applications[1–6]. A THz PCS is composed of two metal pads contacting a semiconductor channel (or a gap, as it is often referred to in the literature), typically composed of GaAs, InP, or silicon with high defect density. When operated as an emitter, a PCS is voltage-biased and illuminated with ultrafast femtosecond laser pulses. Electron–hole pairs are excited in the channel and separated by the bias field, creating a transient current. The transient current decays rapidly due to the sub-picosecond relaxation of excited carriers in the heavily defective semiconductor channel. The transient current in PCS couples with a THz antenna to emit the THz signal to the far field. While substrates with longer carrier lifetime (such as semi-insulating GaAs) can be used for efficient broadband THz emitter through clever engineering (such as the use of interdigitated antenna)[7], short carrier lifetimes

(picosecond) are in general preferred for efficient THz generation when the PCS is coupled with transmission lines/waveguides for THz modulation, studies of device transient phenomena, or almost any other application employing confined fields. The early work on PCS showed very low conversion efficiency for converting pump intensity to THz field[8,9]. The reason is a tradeoff between speed and amplitude: to enable sub-picosecond carrier lifetime, the channel materials are made defect-rich with either low-temperature growth or ion implantation. As a result of increased scattering, the carrier mobility drops substantially[10–12], typically to around 1–200 cm$^2$ V$^{-1}$ s$^{-1}$. The low mobility prevents the excited carriers from traveling a long distance before recombination, leading to a small transient current and weak THz emission. To mitigate this problem, researchers employ interdigitated[13–15] and nanostructured plasmonic[16–19] metal contacts. These designs reduce the carrier transport path length and increase absorption near the contact, making the carrier separation more

[1]Department of Electrical Engineering and Computer Science, University of Michigan, Ann Arbor, MI 48109, USA. ✉e-mail: tnorris@umich.edu; zzhong@umich.edu

efficient. However, they all require substantial surface patterning with sub-micrometer feature sizes, increasing the fabrication complexity.

THz emission from alternative materials such as graphene has also been explored. Graphene is a two-dimensional semi-metal with a very high mobility, exceeding 140,000 cm$^2$V$^{-1}$s$^{-1}$ when suspended and 60,000 cm$^2$V$^{-1}$s$^{-1}$ when encapsulated by hBN at ambient temperature[20,21]. It has been shown that hot carriers in graphene relax within a few picoseconds[22–24], orders of magnitude faster than in conventional defect-free semiconductors. These features make graphene an exciting platform to study ultrafast hot carrier dynamics and THz generation. THz emission has been reported on graphene devices with a suspended graphene–metal interface due to built-in electric fields and photo-thermoelectric effects[25]. Other work has shown THz emission due to photoconductive effects[26] and thermally excited plasmons[27]. However, all these approaches showed a rather low efficiency due to the low optical absorption (around 2.3%[28] in graphene, which is significantly dwarfed when compared to bulk semiconductors.

Even though graphene alone does not emit strong THz radiation, its combination of high mobility and short hot carrier lifetime offers a unique opportunity for functioning as a fast lane for hot carrier transport. In this work, we propose a strategy for high-power THz PCS by combining the strong absorption in bulk semiconductors with a graphene layer as a hot carrier fast lane. Figure 1a illustrates the graphene–silicon hybrid PCS device structure and design strategy for THz generation enhancement. A layer of graphene is placed over a standard silicon Auston switch. When light illuminates the sample, photoexcited carriers are generated mostly in the silicon layer. Instead of inefficient carrier extraction within the silicon, the hot carriers are first transferred to the upper graphene layer, and then the lateral applied electric field enables fast transport of the carriers to the contacts. As graphene has much larger mobility than damaged semiconductors, a significantly larger population of carriers is separated and extracted before recombination, resulting in a stronger transient dipole and subsequent THz generation. This is why we call the parallel graphene channel a "fast lane". The graphene–silicon hybrid device shows up to 80-time enhancement in amplitude compared to the graphene-free silicon PCS, without sacrificing any bandwidth or signal–noise ratio (SNR). Our result sheds light on the engineering of ultrafast hot carrier dynamics in such hybrid systems to facilitate applications in the THz domain. More generally, it works as a good example of complementing 2D material properties with 3D ones for a better device performance unparalleled by either of the materials alone.

## Device fabrication and characterization

Figure 1b shows the graphene–silicon hybrid PCS device structure and illustrates the on-chip pump–probe measurement set-up. The hybrid PCS emitter and a second silicon-only Auston switch (functioning as THz detector) were fabricated on the same die and connected via

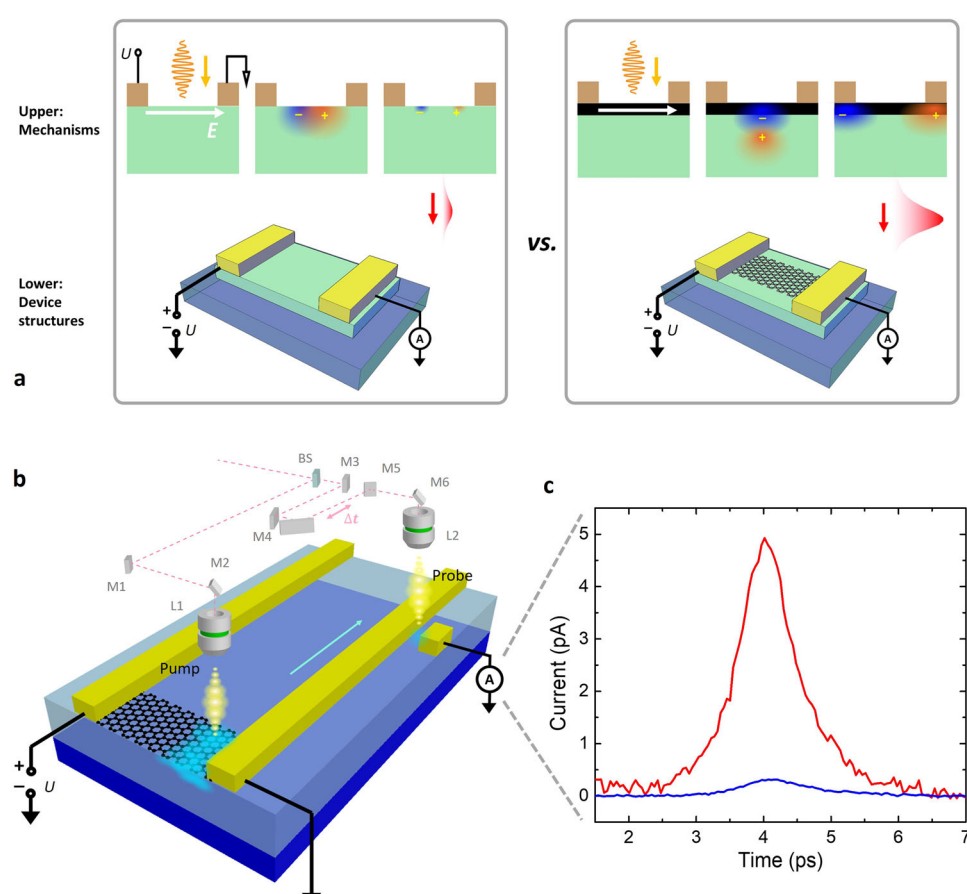

**Fig. 1 | Schematic illustration of the THz emitter with graphene as the hot carrier fast lane. a** Structure of the conventional PCS without graphene (left box, lower panel) versus the device with graphene (right box, lower panel). The corresponding THz field generation mechanism is shown in the upper panels. Hot carriers are majorly generated in O + ion-implanted silicon (green). Hot carriers separate more efficiently in the graphene layer (black) than in the silicon layer, hence creating a stronger THz field on the right. **b** The on-chip pump–probe measurement setup. The beam splitter (BS) separates the input beam into pump and probe beams. A motorized stage controls the time delay (mirror M4) of the two beams, which are focused to hit the sample at the emitter (through L1) and the detector (through L2) respectively. A transmission line is used to couple the field to an Auston switch (detector). **c** Strongly enhanced THz field observed from our hybrid device (red) over graphene-free device (navy). The channel bias is 6 V, with a pump power of 3 mW and probe power of 10 mW for both emitters.

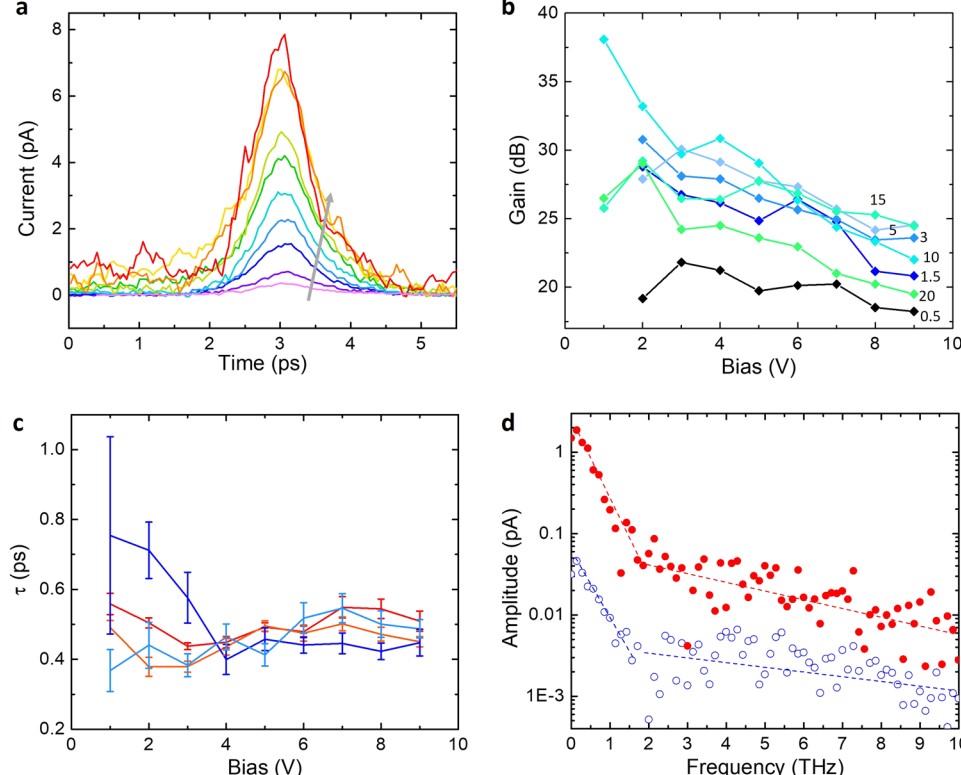

**Fig. 2 | Measured field enhancement and data analysis. a** Time-resolved measurement of received field amplitude under different channel bias, with pump power of 3 mW and probe power of 10 mW. Violet to red (following the gray arrow): channel bias = 1 to 9 V with a 1 V stepped increase. Pink: graphene-free device at 9 V. **b** Gain of generation (measured in power) compared to graphene-free device at identical test conditions. The power gain varies from 19 to 38 dB at different channel bias and pump power (pump powers marked in the graph, unit: mW).

**c** Extracted rise and fall times of the pulses with $\tau$ value from least-square-regression Gaussian fitting. Red: rise time, graphene on silicon device; orange: fall time, graphene on silicon device; navy: rise time, silicon device; blue: fall time, silicon device. **d** Fourier-transformed THz spectrum of graphene device (red dot) and graphene-free device (blue circle). The data in **c** and **d** are extracted from **a**, which shows no decrease in bandwidth or SNR.

metal waveguides (see more details in "Methods" and Supplementary Information). Notably, adding graphene does not necessarily much increase the fabrication complexity, as mature wafer-scale graphene growth[29,30], transfer[31,32], and device fabrication[33–35] have been reported.

During the on-chip pump–probe measurement, a channel bias voltage (between 1 and 9 V) is applied across the graphene channel. A higher voltage up to 45–105 V is possible given the high breaking field of graphene with optimized sample preparation[36,37], so that graphene's high-voltage tolerance does not set a fundamental limitation of the proposed potential applications. A pump laser pulse (Ti:sapphire, mode-locked at 800 nm, repetition rate = 76 MHz, pulse duration <100 fs,) illuminates the metal–graphene–silicon heterostructure. The ultrafast hot carrier transport generates a THz field which is then coupled to the coplanar waveguide. After 100 μm propagation, the signal reaches the second Auston switch functioning as the detector. The detector switch is opened by a time-delayed probe pulse to enable sampling of the transient THz field. We tune the time delay of the pump and probe (or "gate") beams via a motorized delay line (see Supplementary Fig. 1). A chopper modulates the pump beam at 2500 Hz, and the electrical signal is captured at this frequency by a lock-in amplifier for improved SNR. The direct photocurrent measurement on the same device can be found in Supplementary Fig. 3.

## Results

We first directly compare the performance between our graphene–silicon hybrid PCS and silicon PCS with identical design except for the top graphene layer. Figure 1c shows the THz pulse amplitudes measured at the Auston switch detector, with 6 V channel

bias, 3 mW pump power, and 10 mW probe power for both emitters. It is clear that the graphene–silicon hybrid PCS shows a much stronger THz generation. We further studied the THz field amplitudes for our hybrid PCS under different bias voltages across the channel (Fig. 2a). The amplitudes increase linearly with the applied bias voltages (fittings in Supplementary Fig. 2a). The power gains of hybrid PCS over silicon PCS are also plotted with different biases and pump powers (Fig. 2b). We observe a maximum enhancement of 80 times in amplitude (38 dB gain in power, as calculated from the amplitude enhancement) at a low bias and high power, and typical amplitude enhancement of lower 10 s under high bias.

We then measured the bandwidth from hybrid PCS with both the width of fitted Gaussian peaks (Fig. 2c) and Fourier transform of the peaks (Fig. 2d). The detected waveform has a Gaussian shape, instead of being bipolar in many observations, because the signal is confined to a transmission line near the source[38]. We derived the values by fitting the rise slopes and fall slopes in Fig. 2a respectively with Gaussian curves (with $\tau$ values fitted with Gaussian functions, see Supplementary Fig. 2d) using least square linear regression. The extracted values lie around 0.4 ps, which is close to the $\tau$ values extracted using the same method from silicon PCS. Importantly, we observe no bandwidth decrease in the hybrid PCS when compared to silicon PCS. Another important metric for THz emitters is the SNR. Despite the ~10 times increase of the noise level (Fig. 2d) due to graphene's lower resistance, the amplitude increases from 0.04 to 2 pA in the hybrid PCS, leading to a factor of 25 enhancement in SNR. Taken together, the adoption of graphene layer in the THz PCS achieves a large enhancement in THz generation without sacrificing either bandwidth or SNR.

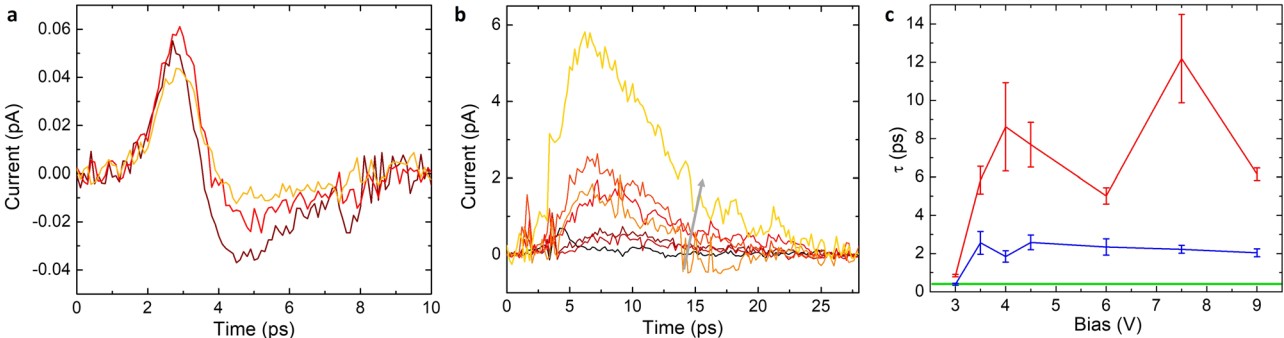

**Fig. 3 | Experimental data for field generation mechanism analysis. a** THz field from a pure graphene device under different channel bias (brown: 0 V; red: 4 V; orange: 8 V), with pump power = 5 mW and probe power = 30 mW. **b** THz field from the same 2D–3D heterostructure with a lightly implanted silicon substrate. Black to yellow: channel bias from 3 V to 9 V with 1 V step increase (gray arrow). **c** Extracted rise (navy) and fall (red) time through Gaussian fit. A comparison is made with the heavily implanted ones (green banner).

Next, we investigated the possible mechanisms contributing to the large enhancement of THz generation in the graphene–silicon hybrid PCS. The generated THz field's amplitude linearly increases for larger channel bias (Supplementary Fig. 2b). The observation eliminates the possibilities of edge effects as the dominating mechanism, such as photo-thermoelectric[39–41] and local built-in field effects.

To further verify the roles played by silicon and graphene in the THz generation process in the 2D–3D heterostructure, we fabricated silicon-free, pure graphene emitters. We illuminated the device at the metal–graphene edge with a different bias applied across the channel and similarly measured the generated pulses using the second Auston switch detector. The signal obtained at the detector also showed picosecond pulses, but importantly, the pulse amplitudes were two orders of magnitude smaller than the hybrid PCS, even though the control device was pumped and probed at a higher power (Fig. 3a).

Another sample was fabricated with an identical structure as the standard device, except for an initial step of ion implantation at a lower flux of $10^{13}\,cm^{-2}$. The lower density of $O^+$ ion-induced defects leads to a longer carrier lifetime in silicon, while the hot carrier lifetime in graphene remains mostly unchanged. Additionally, we performed a second $O^+$ ion implantation to the detector region only. The total implantation flux in the detector region is identical to the heavily implanted samples so that the detector's temporal resolution and sensitivity are identical to those used in other samples. Figure 3b, c suggests that the generated THz field has similar amplitudes but significantly longer rise and fall times determined by the carrier lifetime change in silicon. The observations indicate the THz generation is dominated by the absorption of pump pulses by the silicon layer rather than the graphene layer, followed by some ultrafast process between silicon and graphene, which is to be explored.

Next, we examine how the graphene layer interacts with the silicon layer to enhance field generation. While the charge transfer (CT) model, as depicted in Fig. 1a, seems a plausible explanation of the phenomenon, there are other possible mechanisms. For example, immediately after the carrier excitation in silicon, there may be non-radiative energy transfer (NRET) between the silicon and graphene layer, which might also couple significant energy into hot carriers in graphene. The NRET mechanism has previously been shown to dominate in a diamond-graphene heterostructure[42]. Another possibility is that the insertion of graphene changes the RCL of the circuit, enabling more efficient THz coupling into the waveguide.

To differentiate these mechanisms, we fabricated a graphene top gate (with ALD-grown $Al_2O_3$ as gate dielectrics) on top of the hybrid PCS structure (see Supplementary Information Section I), in order to electrostatically modulate the 2D-3D heterojunction bands. Figure 4a shows the $I–V_g$ transfer curve and the schematic of a typical device.

Unlike a typical symmetric transfer curve centered around the charge neutral point, the electron branch (n-branch) current at the more positive gate bias ($V_g$) is suppressed heavily. Similar behavior was observed on all 8 devices characterized (see Supplementary Fig. 4). We further probed the photocurrent of the device at zero channel bias ($U$) and sweeping the gate bias and the beam spot position along the channel (Fig. 4a inset and Fig. 4b). The result indicates a sign flip of photocurrent when sweeping $V_g$ from negative to positive at regions sufficiently far from the metal edges (indicated as a pink dotted line in Fig. 4b). This indicates inversed vertical built-in field at the graphene–silicon interface under various gate bias.

The transfer curve behavior and the reverse of photocurrent can be understood using band diagrams shown in Fig. 4c. At negative gate bias, the graphene layer is p-doped and depletes the underlying silicon layer. Hence the doping of graphene is well controlled by the top gate. Under positive gate bias, the graphene layer becomes n-doped by the top gate. However, the band distortion caused by this doping drags the silicon layer into an accumulation of electrons, thus partially compensating the top gate tuning of graphene channel conductance. These proposed band diagrams are consistent with our observations and previous reports of $O^+$ ion in silicon as a donor[43]. We further verified the hypothesis by performing $C–V$ measurements between the graphene gate and channel (see Supplementary Fig. 5). The total capacitance increases under positive gate bias, which is consistent with our proposed model, where the parasitic graphene-silicon capacitance abruptly increases at accumulation.

We further studied the gate dependence of the THz amplitude from the hybrid PCS. Figure 4d shows the field's amplitude captured by the X and Y channels of a lock-in amplifier, which reflects both the absolute value and the phase of the signal. The amplitude's absolute values peak at around −3 and 7 V (orange dotted lines), but dips around 3 V (green dotted line), leading to a pair of M-shaped dependence (the red M positioned upward, and the blue M mirrored by the horizontal axis due to the phase of Y channel). These unique gate-dependence results exclude both NRET and enhanced impedance matching as the major contributor to the enhanced THz field in hybrid PCS. NRET happens via dipole-dipole interactions, which depends on the dielectric constant of the environment[44]. The electrostatic gating is too weak to fully suppress the generated THz field at the dip around 3 V. For the hypothesis of enhanced coupling to the waveguide, the field amplitude should follow a monotonic trend similar to that of the channel's transfer curve, which clearly contradicts our experimental results. On the contrary, the proposed CT process agrees with the gate dependence results. As discussed above and shown in the band alignment in Fig. 4d, the sample would reach a flat band condition at small gate bias, where the CT process is suppressed. This explains the dip in THz amplitude observed at 3 V. For

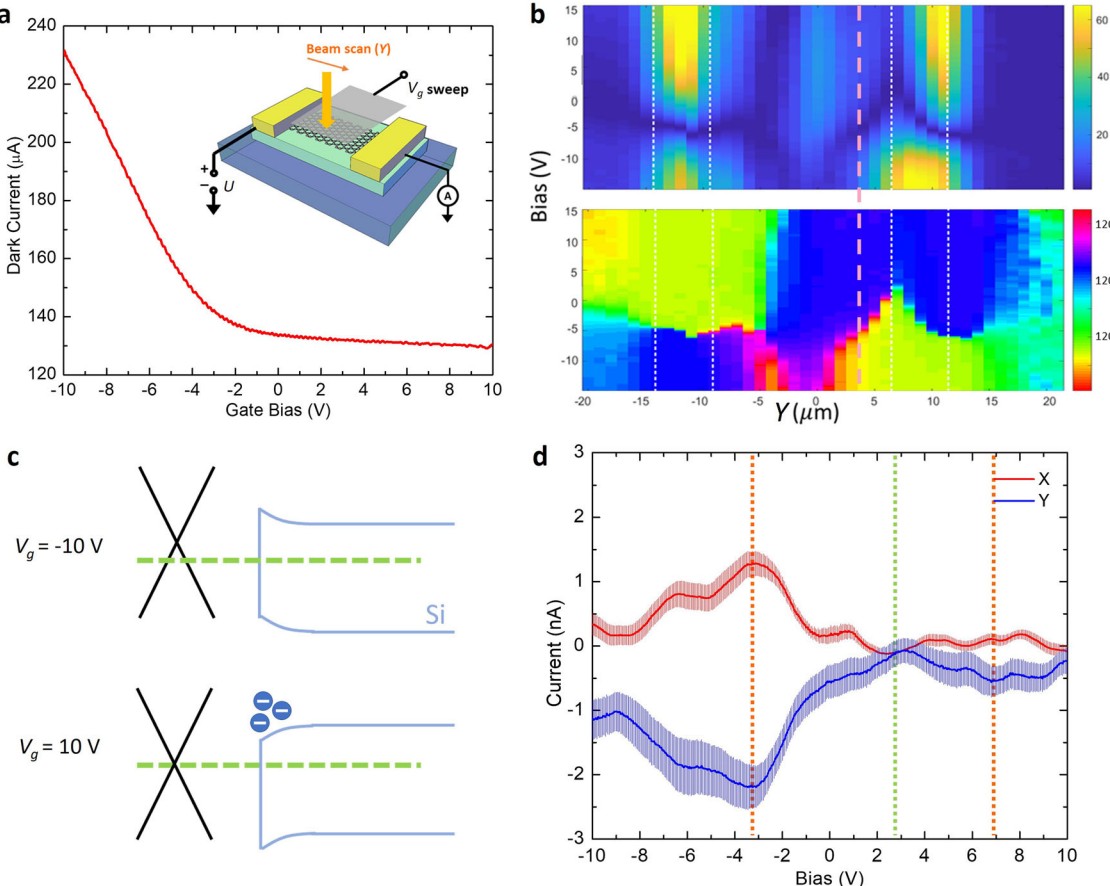

**Fig. 4 | Gate dependence of device performance. a** Current across the device under 1 V channel bias ($U$) and sweeping gate bias ($V_g$). Inset: electrical operation for dark current in **a** and scanning beam measurements in **b**. **b** Spatial scanned pump beam is used to probe the photocurrent at $U = 0$ V. The amplitude (upper) and phase (lower) are plotted with respect to both position and gate bias. **c** The proposed band alignment of the device, with silicon surface depleted at negative gate bias and accumulation at positive gate bias. At large positive $V_g$, the band distortion to silicon electron accumulation counter-dopes the graphene layer. **d** Gate dependence of the peak THz pulse amplitude from the device, with error bars (determined by the standard deviations) and signals from the lock-in amplifier's X (red) and Y (blue) channels, indicating a stable phase of the THz field.

highly positive and negative gate biases, the graphene layer has a lower mobility due to increased carrier-carrier interactions[20,45]. Hence a weaker dipole moment is generated within the shorter hot carrier lifetime. Overall, the picture of CT[46] as the dominate interlayer coupling mechanism is consistent with our proposed hybrid PCS design with highly enhanced THz generation.

Finally, our experimental data suggest that the graphene carrier fast lane is indeed carrying hot carriers during the THz generation. Consider two measurements with different photocarrier lifetime in the silicon layer, i.e., Figs. 2a, 3b. The THz field amplitude decreases slightly from 8 to 6 pA at 9-V channel bias. Suppose the interlayer charge transfer time is significantly slower than the shortest photocarrier lifetime (0.7 ps); then the majority of the photocarriers contributing to Fig. 2a will recombine before transferring to graphene. Hence, the amplitude in Fig. 2a should be much lower than in Fig. 3b, which contradicts our observation. Consequently, the CT process is fast enough to transfer hot carriers to graphene, making the graphene layer a hot carrier fast lane.

## Discussion

By engineering a hot carrier fast lane in graphene–silicon hybrid PCS, we demonstrated significant enhancement of THz field amplitude without sacrificing bandwidth or SNR. Our investigation supports that photocarrier generation mainly happens in the silicon layer; hot carriers are then transferred from silicon to the graphene layer via CT mechanism and use graphene as the hot carrier fast lane to the metal

contacts. Despite the earlier envisions of weak THz emission in graphene-based devices[47], our work shows that a combination of graphene and traditional bulk materials can surpass both materials as an efficient hybrid THz generator. Further THz enhancement may be readily achieved through various approaches, including improvements in graphene quality; replacing silicon with other semiconductors such as low-temperature GaAs, which has a faster relaxation and higher mobility[6,48]; and incorporation of interdigitated electrodes and plasmonic nanostructures. Further developments on similar 2D-3D hybrid systems should lead to a new promising platform for high-performance THz devices and applications. The most direct implications of our findings are for THz generation enhancement in waveguide (essentially, on-chip) geometries. Ultrafast measurements on nanoelectronic devices, ballistic conductors, high-speed on-chip switching devices, etc. are often best-performed on-chip.

## Methods

The device was fabricated on a silicon-on-sapphire substrate, with a silicon thickness of 300 nm. The silicon was first damaged by ion implantation with $O^+$ ions at 100 keV with a flux of $10^{15}$ cm$^{-2}$. The ion-induced defects work as recombination centers. Previous studies reported that the excited carrier lifetime reduces to around 0.7 ps[49,50] at our implantation flux, as verified by time-resolved reflectivity measurements. A CVD-grown graphene layer is then wet-transferred to the substrate and patterned into aμ graphene channel (15 μm long, 10 μm wide) using photolithography and oxygen plasma etching. Finally, a

300-nm-thick layer of gold is deposited onto the sample to form both metal contacts and double-stripe THz waveguides[45] using standard lithography and lift-off processes.

The lifetime of transient carriers in silicon Auston switch is characterized by the same on-chip pump–probe measurement with an identical layout to our main measurements, with the only difference being that the emitter does not have a graphene layer on top. Supplementary Fig. 2c shows the pulse generated by the simple silicon emitter. The observed pulse has a FWHM of 0.9 ps, corresponding to a carrier lifetime of 0.65 ps by deconvolving the emitter and detector transients assuming exponential decays. The value matches well with the value (0.7 ps) observed in samples prepared with identical conditions in the previous literature[50].

## Data availability
The data that support the findings of this study are available from the corresponding authors upon reasonable request.

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

## Acknowledgements

The authors gratefully acknowledge financial support from National Science Foundation grants (ECCS-1509354) and the W. M. Keck Foundation. Devices were fabricated in the Lurie Nanofabrication Facility at the University of Michigan, a member of the National Nanotechnology Infrastructure Network funded by the National Science Foundation.

## Author contributions

D.Z., Z.X., T.B.N., and Z.Z. conceived the experiments. D.Z., Z.L., A.R.G., and W.Z. fabricated the devices. Z.X. and G.C. built the optical set-up. D.Z. and Z.X. performed the on-chip pump–probe measurements. All authors discussed the results and co-wrote the manuscript.

## Competing interests

The authors declare no competing interests.
