## [Peer Review File · Nature Communications]

Strongly Enhanced THz Generation Enabled by a Graphene Hot-Carrier Fast LaneReviewer #1 (Remarks to the Author):

The authors demonstrate a modality of THz emitter photoconductor emitter chip based on graphene integrated on to O+ implanted silicon. The main evidenced claim is that the graphene acts as a "hot carrier fast lane" in the emission process, which is, I think novel, and potentially very noteworthy.

The work appears to be of substantial significance in the area of THz emission and to the aligned area of THz TDS, since it shows how overlaid graphene can generically be used to improve the performance of THz photoconductive switches. I am impressed but the thorough way that the authors substantiate the claimed mechanisms of improvement via hot carrier dynamics, using a series of control experiments to validate their claims. The data analysis as presented looks generally sound, and does the physical interpretations as above, and therefore the conclusions on the emission processes look robust. I would, however, have liked to see additional data provided where the authors speculate that the same effect could be seen with LT-GaAs (line 201) in the conclusions. Such semiconductors are readily available, so why not demonstrate that the emission is so improved in that case too? Also, the notion that the carriers see a "longer travel distance" in LT-GaAs is too colloquial – do the authors mean one of the carrier scattering times, and if so please elucidate this with precision. There are also a few typos in the form of missing references ("bookmark not defined"). The methodology in terms of experimental measurements looks complete and meets the expectations of work within THz-TDS.

In general the level of detail given is fine to allow the data to be reproduced, taking into account the extra information provided in the supplementary information. I would like, however, to see some evidence (via Raman spectroscopy, for example) that the claim to be using graphene (rather than multilayer carbon sheets) can be backed up.

Reviewer #2 (Remarks to the Author):

The authors report a graphene-on-silicon based device for the emission of terahertz radiation. The topic is interesting and appropriate for Nature Communications. The idea of combining the high-mobility of graphene and the strong optical absorption of silicon for a photoconductive emitter is no doubt interesting and worthy of investigation. The English language used in the manuscript is appropriate for scholar publication. However, there are serious concerns with the study as described in this manuscript, which I detail in the following paragraphs, and therefore I can not recommend its publication in the current form.

1. From a general perspective the main problem I found is that there is no clear demonstration of the capacity of this device to produce a reasonable amount of THz power that can be coupled out of the device. In this sense, there are two aspects that concern me. Firstly, the study does not demonstrate actual emission of terahertz radiation in the far field, which would be the real motivation for it. In other words, the emitter and detector lie on the same substrate within 100um, which is a fraction of a wavelength, from each other, therefore this does not show that actual radiation is coupled out of the device, and the effect itself might be only a capacitive effect in this small device. Secondly, there is no clear indication that the device can sustain a large enough bias voltage to produce a significant emission. A dark resistivity measurement is not presented which would demonstrate that for instance ~10-25V across a 10-20um gap can be tolerated by the device, which are typical values for THz emitters currently used, with just 3V or so, the amount of power emitted will actually be lower than that of a SI-GaAs emitter with a bias of 10 to 25V. A comparative measurement with a standard emitter would be desirable in order to be able to make claims about the actual improvement.

2. While very short carrier lifetime is indeed required for good photoconductive detectors, this is in general not true for photoconductive emitters. For emitters the contrast between dark and bright conductivity and the possibility of high bias voltage are the main characteristic that determines its output power. This is wrong all along the narrative in the current manuscript. The "short" carrier lifetime is needed in the case of silicon because the lifetime in non-damaged-silicon is so long that the conductivity remains almost unchanged for hundreds of nanoseconds, which means that the repetition period of the laser (in the several tens of MHz) is shorter than the lifetime, and the silicon remains conductive for the entire duration of the laser repetition cycle (~10ns).

3. The power comparison of the graphene vs the non-graphene samples is more or less fair, this is not the case when comparing the devices with different implantation states, since the implantation will heavily influence the performance of the detector, which in this case is on the same substrate, those comparisons are therefore inappropriate/unfair.

4. The argument presented between lines 101 and 111 is hard to follow. the RCL model is referred to without any explanation of what RCL circuit was assumed, and how that is linked to the actual device presented here.

5. Most of the discussion presented between line 112 and 130 does not make much sense. Furthermore, the higher emission when exciting a photoconductive emitter near the anode is very well documented experimentally [Appl. Phys. Lett. 59,1972 (1991); IEEE J. Quantum Electron. 32, 1664 (1996)], and explained theoretically [Phys. Rev. B 71, 195301 (2005)].
1972 (1991)

4. The authors claim a lifetime of their substrates of 0.7ps (line 78). How did they determine this? This is actually critical for the performance of the detector.

5. The size of the gap between contacts is never described in the manuscript, and it is a very important parameter, I estimated it to be of the order of 100um by looking at the schematics and the non-scaled photograph in

6. The discussion presented between lines 147 and 166 was extremely confusing to me. The authors refer to "the gate" without defining it, then the voltage V_g is mentioned, but it is not clear if this is the same voltage that is referred to as U in Fig.1. The fact that there is a current at 0-voltage (shown in Fig.4a) does not make much sense. All-in-all I believe this paragraph was written without much care, and is confusing, I must confess that after reading it 3 times I could not make any sense out of it.

7. Between lines 167 and 169, a description of what is seen in Fig 4d is presented, but this does not match what is seen in the figure. For instance the authors discuss voltages between -3V and +7V, but in the figure only positive voltages are shown. Furthermore, the minimum of the "M"-curve appears at 3V for one of the two curves, but not for the other. All in all this part was also confusing, and there are inconsistencies between the text and the plot.

8. The paragraph between lines 183 and 190, also make claims comparing the results in figure 2 and 3, however, for the reasons already mentioned, about the detector being strongly affected by the carrier lifetime, it is not possible to make fair comparisons.

9. Other specific comments:

a. line 12: "To obtain picosecond ultrafast pulses, semiconductors are typically heavily damaged to reduce the carrier lifetime". Please reconsider

b. line 28: "channel", please consider using the more common term "gap"

c. line 30: idem

d. lines 30-32: please reconsider the sentence starting "The transient current ..."

e. line 33: the authors talk about "low quantum efficiency", yet this is not a term that is

normally used at least in the THz community, it would be easy to find efficiencies that are not that low since THz photons contain about 3 to 4 orders of magnitude less energy than IR-VIS photons, one could have "quantum efficiencies" close to 1 and still convert 0.1% to 0.01% of the laser energy into terahertz energy. Perhaps "low conversion efficiency" is more appropriate, although in general this is also not a parameter that is typically reported, as the bias voltage also plays a fundamental role in the conversion, so it is not a property of the device as such.

f. line 35: "carrier recombination", I believe the authors refer to carrier trapping, since defects act as traps, but this is a different process to the recombination.

g. line 36-37: the carrier mobility drops, but not by "a few orders of magnitude" please reconsider

h. line 42: "reducing the overall device active area" This is actually not entirely true, the patterned antennas actually allow larger laser beams, which results in a larger emitter area, in the case of plasmonic contacts, they act a bit like "light funnels" so in the end the effective area is not necessarily reduced.

i. line 73. Although "pump-probe" has been wrongly used by many authors, in general the THz community understands as a pump-probe setup the one where a sample (not the emitter or detector) is optically pumped by a third laser pulse, and probed by the THz pulse. The more accepted, and less ambiguous, term used now is THz-generation (beam) and gate (detector beam).

j. The dimensions of the device are not mentioned in the manuscript, please include them

k. The repetition rate of the laser and power of each beam are also not mentioned in the device fabrication and characterization section

l. Your pulses look indeed Gaussian, but the typical shape of THz waveform pulses is bipolar, or even more complicated, the pulses should integrate zero over time, this is for charge conservation reasons. Perhaps your current signals require further processing, I believe this is related to the carrier lifetime not being short enough in the detector.

m. line 128: there is an error (I think a reference)

n. line 179: idem.

Reviewer #3 (Remarks to the Author):

In general, my recommendations are that the authors consider the comments below and re-submit the paper, potentially dividing it into two separate papers (one focusing on fully quantifying the extent of the enhancement, the other investigating potential mechanisms, see comment 9). It's clear that much work went into the results of this paper, and there is significant room for increased clarification on the details of and interpretation of the results.

Major Revisions

Comment 1

I don't think the phrasing "Terahertz Emission" in the title is appropriate. Typically, "emission" indicates that a propagating wave is generated and characterized by propagating the wave a significant distance and coupling into a separate device. Here, the generation and characterization all take place on the same device. I agree that the results indicate that a stronger pulse is generated with your device, so maybe that would be a better word for the title (and phrasing throughout the paper), but I hesitate to consider what is being done here as terahertz emission. I would recommend reconsidering this or reviewing the literature to see if this phrasing is commonly used for similar experimental configurations.

Comment 2

Line 99 and Associated Figures: "Gain" either needs to be mathematically defined, or (what I would really recommend) replaced with simple with "enhancement" as a direct multiplier (like you state with the 80 times greater power). Also, is this truly a calculation of power, or are you calculating enhancement/gain based on the emission

amplitude value? If it is not a calculation of power (if it is, please define) then that needs to be restated. Gain can mean different things to different people, so I really think to avoid confusion simply stating as an emission amplification multiplier would be best.

Comment 3

Line 101: Please provide more explanation/theory/background on what σ is and how it is derived from the curves. I'm not sure if σ is standard notation for this value, but when I see σ I typically think electrical conductivity. If it is not standard, and is simply the notation you decided on, I would recommend using a different notation, maybe τ since that is typically associated with time constants.

Line 105: The reference to the extracted values from silicon PCS is ambiguous, what extracted values are you referring to? Are these values you measured, or known material properties?

Comment 4

Is there really anything special about graphene as a material choice, or could the same effect be achieved using any transparent material film that increases the surface conductivity? For example, if this were reproduced using a layer of ITO that had a similar conductivity in place of the graphene, would similar results be expected?

Comment 5

The bias voltages examined in this paper seem low. Although the electrode spacing isn't explicitly stated, I believe a device such as this should be able to be operated in the range of several 10s of volts, an order of magnitude higher. I can see how the graphene could significantly reduce this maximum bias voltage. To clarify this, please include results and discussion on the maximum operating bias voltage for both the standard and graphene enhanced devices. Since, from an emitter perspective, operating at lower biases isn't really advantageous, a true comparison of benefit of the graphene device would be to compare the pulse amplitude (and possibly SNR) at the highest operable bias with the standard device at its own highest operable bias (which I would believe to be much higher). If it turns out that when operated at each maximum operating bias the graphene based device does not significantly perform better, it is still possible that a device such as this could have advantages as a detector, where the THz induced bias field is much lower. This may be something to consider and comment on in the paper.

Comment 6

Line 115-118: It's not clear how showing strongest THz generation when excitation occurs near the electrode eliminates the possibility of edge effects dominating. If I'm following, wouldn't higher emission when the edge is illuminated with the pump support the possibility of edge effects? Please elaborate on these claims.

Comment 7

Lines 119-121: Please concisely state up front what is explicitly being tested by studying these two special cases of the device. My understanding is that the motivation for the graphene only device is to remove any photoconductive effect from the substrate, but I'm not understanding how the examination of the low O implanted device supports the claim that THz emission is dominated by absorption in the Si layer.

Comment 8

Lines 155-156: The claim "this is a clear sign of..." Is not clear to me at all. I'm just not following the logical flow from the rest of the paragraph, and I'm not sure what this claim is trying to state. Please revise and clarify.

Comment 9

In general, the final 8 paragraphs of the results section (lines 112-202) are unclear and difficult to follow. It's clear the authors performed a lot of work, but I think it is trying to be condensed to fit all the results into a single paper.

I would highly recommend the authors consider break this work into two papers, one focusing on characterizing the extent of the enhancement itself, addressing some of the comments I've made here, and another paper more fully investigating the mechanisms of the enhancement. There seems to be a lot of good work here that could be expanded into two high quality papers, but as it stands now there is a lot of unclear and difficult to follow logic behind the conclusions that are made.

Minor Comments

Line 57: Fig 1 is, in general, not very clear. Some additional text/labels would be useful

to clarify what each part of each image is. Additionally, being that Fig 1a, Fig 1b, and Fig1c are all referenced in different sections of the paper (1a in the Intro, 1b in the Fab and Characterization, 1c in the results) it might flow better breaking the figure up into three different figures.

Line 41 and 78: The claim of nanopatterning to achieve nanoscale plasmonic electrodes is valid, but to someone such as myself who is unfamiliar with graphene processing, wet-transfer of a graphene film also sounds difficult. Is this a more (or less) scalable process than nanolithography? Please provide a reference supporting that it is, or reconsider the claim that there are processing advantages to this type of device over current nanopatterned electrode configurations.

Fig 2 and 3: Please add legends or some other visual indicator to each graph to help clarify what each trace represents. For example, 2a, where there are many curves, something like an upward arrow through the curves and text stating "increasing channel bias" would help.

Fig 4c is not clear, without extensively reading the text and filling in gaps as to what represents what in the illustration. Please add clarification to help the image become more understandable from a "stand-alone" perspective.

Point-by-point Responses to Reviewers' Comments

Reviewers' comments are in **black**; our responses are in **blue**;

Reviewer #1 (Remarks to the Author):

The authors demonstrate a modality of THz emitter photoconductor emitter chip based on graphene integrated on to O+ implanted silicon. The main evidenced claim is that the graphene acts as a "hot carrier fast lane" in the emission process, which is, I think novel, and potentially very noteworthy.

The work appears to be of substantial significance in the area of THz emission and to the aligned area of THz TDS, since it shows how overlaid graphene can generically be used to improve the performance of THz photoconductive switches. I am impressed but the thorough way that the authors substantiate the claimed mechanisms of improvement via hot carrier dynamics, using a series of control experiments to validate their claims.

The data analysis as presented looks generally sound, and does the physical interpretations as above, and therefore the conclusions on the emission processes look robust. **I would, however, have liked to see additional data provided where the authors speculate that the same effect could be seen with LT-GaAs (line 201) in the conclusions. Such semiconductors are readily available, so why not demonstrate that the emission is so improved in that case too?**

We thank the referee for the thoughtful comment. We agree that this would be an interesting direction to pursue, as we would expect that the observed THz enhancement enabled by graphene hot carrier fast-lane should exist in other material systems as well. This is something we would like to do (though are presently not able), and we would encourage groups who see our paper to develop the concept in other materials where they have the relevant materials expertise. The focus of this manuscript, however, is on demonstrating a new concept in a readily available and commonly used material platform, and investigations into other materials must remain a subject for future work.

We note further that our group does not presently have prior experience working with low-T GaAs wafers, including etching, doping, metallization, layer transfer, etc., all of which are non-trivial.

We trust that the work presented here is sufficient for a first publication.

Also, the notion that the carriers see a “longer travel distance” in LT-GaAs is too colloquial – do the authors mean one of the carrier scattering times, and if so please elucidate this with precision.

We thank the referee’s comment, and have replaced “longer travel distance” with “higher mobility”, which is discussed in ref [6] (Burford N M, et al. Optical Engineering, 2017, 56(1): 010901.) and ref [47] (Warren A C, et al. Applied physics letters, 1991, 58(14): 1512-1514.)

There are also a few typos in the form of missing references (“bookmark not defined”).

Done. Thanks for the catch.

The methodology in terms of experimental measurements looks complete and meets the expectations of work within THz-TDS.

In general the level of detail given is fine to allow the data to be reproduced, taking into account the extra information provided in the supplementary information.

I would like, however, **to see some evidence (via Raman spectroscopy, for example) that the claim to be using graphene (rather than multilayer carbon sheets) can be backed up.**

We thank the referee’s comments. Our groups have extensive experience working with CVD graphene, both from our own growth and commercially available samples.

Shown below is a typical Raman spectrum obtained on graphene samples from the same batch. The sample is wet transferred to a silicon substrate with the identical procedures used in the device fabrication. The Raman spectrum shows G and 2D peaks of graphene, alongside with other features from the silicon substrate.

Reviewer #2 (Remarks to the Author):

The authors report a graphene-on-silicon based device for the emission of terahertz radiation. The topic is interesting and appropriate for Nature Communications. The idea of combining the high-mobility of graphene and the strong optical absorption of silicon for a photoconductive emitter is no doubt interesting and worthy of investigation. The English language used in the manuscript is appropriate for scholar publication. However, there are serious concerns with the study as described in this manuscript, which I detail in the following paragraphs, and therefore I can not recommend its publication in the current form.

1. From a general perspective the main problem I found is that there is no clear demonstration of the capacity of this device to produce a reasonable amount of THz power that can be coupled out of the device. In this sense, there are two aspects that concern me. Firstly, the study does not demonstrate actual emission of terahertz radiation in the far field, which would be the real motivation for it. In other words, the emitter and detector lie on the same substrate within 100 μ m, which is a fraction of a wavelength, from each other, therefore this does not show that actual radiation is coupled out of the device, and the effect itself might be only a capacitive effect in this small device.

We appreciate the referee's comment and correspondingly have changed "terahertz emission" to "terahertz generation" in the title and the rest of the manuscript to better reflect that far-field radiation is not reported in this work.

We do not agree, however, that only far-field radiation is interesting or important for THz generation. There are *many* applications of THz fields confined to waveguides or transmission lines. While there are some important differences between radiation from a photoconductive switch into a guided mode and to free space from an antenna (which will result in different pulse shapes, bandwidth, etc.), the fundamental process of generating a transient photocurrent is essentially the same in both cases. The scope of our research program reported here was confined to transmission line experiments; we would enthusiastically hope that other groups (or our own, subject to future funding support) might be able to further develop the concept we present here for free space applications, but we believe the concept and physics presented here in the context of transmission line applications is fully sufficient for publication.

Finally, we do note that we performed control experiments on Si with the same geometry but without graphene, as well as control experiments done on graphene (without Si), which rule out a capacitive effect.

Secondly, there is no clear indication that the device can sustain a large enough bias voltage to produce a significant emission. A dark resistivity measurement is not presented which would demonstrate that for instance $\sim 10\text{-}25\text{V}$ across a $10\text{-}20\mu\text{m}$ gap can be tolerated by the device, which are typical values for THz emitters currently used, with just 3V or so, the amount of power emitted will actually be lower than that of a Si-GaAs emitter with a bias of 10 to 25V . A comparative measurement with a standard emitter would be desirable in order to be able to make claims about the actual improvement.

Two points should be noted:

(1) We would like to point out that the devices were biased up to 9V instead of the 3V quoted, as shown in Fig.2 and Fig. 3. Because the devices were not passivated and without optimized packaging, and all tests were carried out in air under ambient condition, we intentionally avoided applying a high voltage and did not perform damage testing; this would remain for future work.

(2) All control experiments were conducted under the same biasing and testing conditions; hence we focused on comparing Si-based devices with and without the graphene in direct comparison in order to

understand the relevant device physics. Optimization of power (including maximizing the applied voltage) remains for future work; the physical conclusion regarding enhancement at a given bias is robust. (Note: referee #3 comment #5 raises a similar point; please see our additional discussion there.)

2. While very short carrier lifetime is indeed required for good photoconductive detectors, this is in general not true for photoconductive emitters. For emitters the contrast between dark and bright conductivity and the possibility of high bias voltage are the main characteristic that determines its output power. This is wrong all along the narrative in the current manuscript. The "short" carrier lifetime is needed in the case of silicon because the lifetime in non-damaged-silicon is so long that the conductivity remains almost unchanged for hundreds of nanoseconds, which means that the repetition period of the laser (in the several tens of MHz) is shorter than the lifetime, and the silicon remains conductive for the entire duration of the laser repetition cycle (~10ns).

The referee is correct that, for free-space THz emission, it is possible to use longer-lifetime semiconductors, relying on the fact that the far-field emission is proportional to dJ/dt . Sometimes semi-insulating GaAs is used, for example, although LT-GaAs is also frequently used, where the growth parameters are chosen such that picosecond lifetimes are employed. Applications using Si generally employ damage producing ps lifetimes. Since many THz implementations do employ short-lifetime material, the effect discussed in our work (the "hot carrier fastlane") will be directly relevant to those implementations.

The most important point, however, is that for THz pulses on transmission lines, which is what we are concerned with in this study, ps lifetimes are required. Otherwise the voltage waveform is a step function, which is not useful for THz modulation, studies of device transient phenomena, or almost any other application employing confined fields in waveguides/transmission lines. Hence we focus exclusively on ps-lifetime material in this study.

We note that in the revised manuscript, we have made clear that we are not reporting free-space THz emission. We have also revised the abstract and introduction to confine our discussion of the preference of short carrier lifetime to PCS coupled with transmission lines/waveguides.

3. The power comparison of the graphene vs the non-graphene samples is more or less fair, this is not the case when comparing the devices with different implantation states, since the implantation will heavily influence the performance of the detector, which in this case is on the same substrate, those comparisons are therefore inappropriate/unfair.

We would like to point out that the purpose of the control on Si with different implantation is to confirm that the absorption mainly happens in the silicon and not the graphene; *the Si switch performance is carefully made identical to the other devices for this particular control by running a second implantation only to the detector region.* We further clarify the issue with the following sentences added to the control experiment section:

“Additionally, we performed a second O+ ion implantation to the detector region only. The total implantation flux in the region is identical to the heavily implanted samples, so that the detector’s temporal resolution and sensitivity are identical to those used in other samples.”

4. The argument presented between lines 101 and 111 is hard to follow. the RCL model is referred to without any explanation of what RCL circuit was assumed, and how that is linked to the actual device presented here.

We thank the reviewer for the valuable advice. The key point here is that we observe a strong (25-time) enhancement of SNR, despite a larger noise due to the fact that graphene layer lowers the channel resistance. A lower channel resistance can lead to larger thermal noise, which is supported by the cited RCL modeling reference. However, we did not perform RCL modellings in this work.

We have revised the manuscript to avoid any potential confusion.

5. Most of the discussion presented between line 112 and 130 does not make much sense. Furthermore, the higher emission when exciting a photoconductive emitter near the anode is very well documented experimentally [Appl. Phys. Lett. 59,1972 (1991); IEEE J. Quantum Electron. 32, 1664 (1996)], and explained theoretically [Phys. Rev. B 71, 195301 (2005)].

1972 (1991)

We appreciate the referee's comments. We have revised the paragraph, moved the less straightforward discussions on spatial resolution to the supplementary material, discussed the possibility of edge effects with other stronger evidence, and cited the above references. We do want to emphasize the following:

1. The purpose of this portion of the discussion is to understand the mechanism for the observed strong enhancement between device with and without graphene layer, not the enhancement effect when near anode as has been studied previously in the quoted reference.

2. The PTE effect and built-in field induced photovoltaic effects will have much sharper position dependence than our observation, given the very small beam diameter ($< 2 \mu\text{m}$) and long channel ($15 \mu\text{m}$). Nevertheless, the linear channel-bias dependence and negligible emission under zero bias suggests that edge effects are not dominating the THz field generation.

3. A minor point, the stronger emission near anode is attributed to an uneven field distribution at the reverse-biased contact in the quoted literature. It may be very different given that the graphene/metal junction has a much weaker built-in field (graphene is a semi-metal). Further study on this might be needed, but is outside of the scope of current paper.

4. The authors claim a lifetime of their substrates of 0.7ps (line 78). How did they determine this? This is actually critical for the performance of the detector.

We followed the existing literature, specifically references 28 and 29, to estimate the lifetime from the transient measurement. We used the standard and well-accepted procedure and dose. Beyond a threshold dose the lifetime does not further decrease below 0.7 ps, which is mentioned in the reference. Supplementary Fig. 2c shows the pulse generated by a simple silicon emitter. The observed pulse width is actually longer than the actual detector ON time, considering the dispersion of the THz field during its propagation along the waveguide.

5. The size of the gap between contacts is never described in the manuscript, and it is a very important parameter, I estimated it to be of the order of 100um by looking at the schematics and the non-scaled photograph in

We thank the referee's comment. The gap size is 15 μm . We apologize for any confusion, and have now clearly stated the device geometry in the device fabrication portion.

6. The discussion presented between lines 147 and 166 was extremely confusing to me. The authors refer to "the gate" without defining it, then the voltage V_g is mentioned, but it is not clear if this is the same voltage that is referred to as U in Fig.1. The fact that there is a current at 0-voltage (shown in Fig.4a) does not make much sense. All-in-all I believe this paragraph was written without much care, and is confusing, I must confess that after reading it 3 times I could not make any sense out of it.

We thank the referee's comments and apologize for any confusion. Even though all device structure and biasing info, including bias voltage, gate voltage, were defined either in the main text or the supplementary information, it can be confusing for someone not familiar with a graphene device. We have made effort to clarify that in the revision with both texts and figures.

For the specific questions mentioned above: (1) the gate is referred to the gate electrode fabricated on top of the hybrid PCS structure (see supplementary Section I). We also added a schematic of the gate to Fig. 4a inset in the main manuscript. (2) 0-voltage in Figure 4a means $V_g = 0\text{ V}$ on the gate, not the bias voltage. The x-axis for Fig. 4a is gate voltage.

7. Between lines 167 and 169, a description of what is seen in Fig 4d is presented, but this does not match what is seen in the figure. For instance the authors discuss voltages between -3V and +7V, but in the figure only positive voltages are shown. Furthermore, the minimum of the "M"-curve appears at 3V for one of the two curves, but not for the other. All in all this part was also confusing, and there are inconsistencies between the text and the plot.

We appreciate the referee's comment and have revised both the text and the figure to clarify the observed trend.

"Fig. 4d shows the field's amplitude captured by the X and Y channels of a lock-in amplifier, which reflects both the absolute value and the phase of the signal. The amplitude's absolute values peak at around -3 V and 7 V (orange dotted lines), but dips around 3 V (green dotted line), leading to a pair of M-shaped dependence (the red M positioned upward, and the blue M mirrored by the horizontal axis due to the phase of Y channel). "

8. The paragraph between lines 183 and 190, also make claims comparing the results in figure 2 and 3, however, for the reasons already mentioned, about the detector being strongly affected by the carrier lifetime, it is not possible to make fair comparisons.

As we explain in previous response, the detectors in two devices are implanted with the same dose while the PCS channel is doped differently. Hence the comparison is fair. We hope the revised description is clear to our readers.

9. Other specific comments:

a. line 12: "To obtain picosecond ultrafast pulses, semiconductors are typically heavily damaged to reduce the carrier lifetime". Please reconsider

Unfortunately, the reviewer's comment here is incomplete, probably by mistake, so we are not quite sure how to address it. In response, to this comment, we revised the text from 'heavily damaged' to 'with high defect density'.

b. line 28: "channel", please consider using the more common term "gap"

Thanks for the comment. We feel that using 'gap' and 'channel' has different benefits throughout the paper. The word 'gap' is indeed widely used by convention in many PCS literatures, while our choice of channel better describes our device following the convention of transistors. To avoid confusion, we added the following words to the manuscript:

“(or a gap, as it is often referred to in the literature)”

c. line 30: idemd. lines 30-32: please reconsider the sentence starting "The transient current ..."

This is not a clear comment/critique, so we are not quite sure how to address it.

e. line 33: the authors talk about "low quantum efficiency", yet this is not a term that is normally used at least in the THz community, it would be easy to find efficiencies that are not that low since THz photons contain about 3 to 4 orders of magnitude less energy than IR-VIS photons, one could have "quantum efficiencies" close to 1 and still convert 0.1% to 0.01% of the laser energy into terahertz energy. Perhaps "low conversion efficiency" is more appropriate, although in general this is also not a parameter that is typically reported, as the bias voltage also plays a fundamental role in the conversion, so it is not a property of the device as such.

We agree with the referee that “conversion efficiency” is a more relevant term, and have modified the text accordingly.

f. line 35: "carrier recombination", I believe the authors refer to carrier trapping, since defects act as traps, but this is a different process to the recombination.

We changed ‘carrier recombination’ to ‘carrier lifetime’.

g. line 36-37: the carrier mobility drops, but not by "a few orders of magnitude" please reconsider

We changed it to ‘drops substantially’.

h. line 42: "reducing the overall device active area" This is actually not entirely true, the patterned antennas actually allow larger laser beams, which results in a larger emitter area, in the case of plasmonic contacts, they act a bit like "light funnels" so in the end the effective area is not necessarily reduced.

We removed the words to avoid any controversy.

i. line 73. Although "pump-probe" has been wrongly used by many authors, in general the THz community understands as a pump-probe setup the one where a sample (not the emitter or detector) is optically pumped by a third laser pulse, and probed by the THz pulse. The more accepted, and less ambiguous, term used now is THz-generation (beam) and gate (detector beam).

We understand there are different terminology regarding this within the literature. We do believe our terminology used in this work is fully within the conventions of much of the literature, and is completely unambiguous in the context of our text. In order to remove any possible ambiguity, we have added “or gate” to the text (line 95) to indicate that the probe pulse indeed gates the signal.

j. The dimensions of the device are not mentioned in the manuscript, please include them

Done

k. The repetition rate of the laser and power of each beam are also not mentioned in the device fabrication and characterization section

We’ve added the repetition rate (76 MHz) in the revised manuscript. The pump and probe power are described in the captions of the relevant figures.

I. Your pulses look indeed Gaussian, but the typical shape of THz waveform pulses is bipolar, or even more complicated, the pulses should integrate zero over time, this is for charge conservation reasons. Perhaps your current signals require further processing, I believe this is related to the carrier lifetime not being short enough in the detector.

The referee is correct in his/her observation that the waveform is not bipolar. In fact, it is not required that the pulse be bipolar with an obvious zero time integral within the experimental scan range. It is a common misconception that this is required, but in fact this is only required in the FAR FIELD of a source. (Of course, the integral must be rigorously zero in the far field, but it can indeed be nonzero near the emitter). This subject has been dealt with in great detail in a classic paper by Phil Bucksbaum and his student D. You back in the 1990's; the reference is:

D. You and P. H. Bucksbaum, "Propagation of half-cycle far infrared pulses," J. Opt. Soc. Am. B **14**, 1651-1655 (1997).

That paper shows that THz pulses can appear as quasi-unipolar for quite surprising distances, and this is the case with our experimental pulses. The detection is not fully in the far field, as the signal is confined to a transmission line near the source.

Now, it is true that there must be a negative component that would *eventually* integrate to zero. However, the amplitude of this component can be so small that it is completely within the noise limit of the data, and correspondingly the negative "tail" can be as long as nanoseconds (corresponding essentially to the re-charging of the semiconductor gap).

We have discussed the issue and added this reference in the supplementary, so that any reader may find this, but a long discussion of the pulse shape would not be necessary, as the above reference is sufficient.

m. line 128: there si an error (I think a reference)

Corrected.

n. line 179: idem.

Corrected.

Reviewer #3 (Remarks to the Author):

In general, my recommendations are that the authors consider the comments below and re-submit the paper, potentially dividing it into two separate papers (one focusing on fully quantifying the extent of the enhancement, the other investigating potential mechanisms, see comment 9). It's clear that much work went into the results of this paper, and there is significant room for increased clarification on the details of and interpretation of the results.

Major Revisions

Comment 1

I don't think the phrasing "Terahertz Emission" in the title is appropriate. Typically, "emission" indicates that a propagating wave is generated and characterized by propagating the wave a significant distance and coupling into a separate device. Here, the generation and characterization all take place on the same device. I agree that the results indicate that a stronger pulse is generated with your device, so maybe that would be a better word for the title (and phrasing throughout the paper), but I hesitate to consider what is being done here as terahertz emission. I would recommend reconsidering this or reviewing the literature to see if this phrasing is commonly used for similar experimental configurations.

We thank the referee's comment. We have changed the "Terahertz Emission" to "Terahertz Generation" in the title and the main text to reflect what is experimentally demonstrated in the manuscript.

Comment 2

Line 99 and Associated Figures: "Gain" either needs to be mathematically defined, or (what I would really recommend) replaced with simple with "enhancement" as a direct multiplier (like you state with the 80 times greater power). Also, is this truly a calculation of power, or are you calculating enhancement/gain based on the emission amplitude value? If it is not a calculation of power (if it is,

please define) then that needs to be restated. Gain can mean different things to different people, so I really think to avoid confusion simply stating as an emission amplification multiplier would be best.

We appreciate the referee's comment. We revised the manuscript and emphasized that the gain corresponds to estimation of power. We used enhancement (emphasizing that it is the amplitude) to describe the amplitude increase.

We also clarify that the gain in power is calculated from the amplitude, rather than direct power measurement in the revision.

Comment 3

Line 101: Please provide more explanation/theory/background on what σ is and how it is derived from the curves. I'm not sure if σ is standard notation for this value, but when I see σ I typically think electrical conductivity. If it is not standard, and is simply the notation you decided on, I would recommend using a different notation, maybe τ since that is typically associated with time constants.

Thanks. We revised the main manuscript with the following changes:

1. Replaced σ with τ in the notation.
2. Explained briefly in main manuscript about the Gaussian fittings.
3. Added more detailed explanation, with additional figure and equations in the supplementary.

Line 105: The reference to the extracted values from silicon PCS is ambiguous, what extracted values are you referring to? Are these values you measured, or known material properties?

We revised the sentence as follows:

'which is close to the τ values extracted using the same method from silicon PCS.'

Comment 4

Is there really anything special about graphene as a material choice, or could the same effect be achieved using any transparent material film that increases the surface conductivity? For example, if this

were reproduced using a layer of ITO that had a similar conductivity in place of the graphene, would similar results be expected?

We thank the referee of the thoughtful comment. To function as a hot carrier transport fast-lane, the material requires: 1) efficient charge transfer from silicon to the conducting channel serving as the fast-lane; 2) high carrier mobility such that the carrier can be collected at the contact rather than recombine within the channel or be trapped back into silicon layer. Graphene is probably the natural choice that satisfy these requirements, but it is certainly highly valuable to explore other material options.

In this view, a simple enhancement of conductivity is insufficient. In fact, if a transparent conductor with high conductivity, such as ITO, is placed across the channel (gap), then the gap essentially shorts out the applied bias. Graphene is a material that provides high mobility for energetic electrons, but when the system is not illuminated the actual conductivity is low (as the carrier density is near zero), and a significant bias voltage can be applied. Other materials with high mobility for energetic electrons may be possible, but the off-state conductance must be low.

Comment 5

The bias voltages examined in this paper seem low. Although the electrode spacing isn't explicitly stated, I believe a device such as this should be able to be operated in the range of several 10s of volts, an order of magnitude higher. I can see how the graphene could significantly reduce this maximum bias voltage. To clarify this, please include results and discussion on the maximum operating bias voltage for both the standard and graphene enhanced devices. Since, from an emitter perspective, operating at lower biases isn't really advantageous, a true comparison of benefit of the graphene device would be to compare the pulse amplitude (and possibly SNR) at the highest operable bias with the standard device at its own highest operable bias (which I would believe to be much higher). If it turns out that when operated at each maximum operating bias the graphene based device does not significantly perform better, it is still possible that a device such as this could have

advantages as a detector, where the THz induced bias field is much lower. This may be something to consider and comment on in the paper.

The reviewer raises an interesting point, which is in fact along similar lines to the comment of reviewer #1 regarding the magnitude of the bias voltage, so we partly refer to our reply above. We add a couple of additional comments here.

First, our goal in this work is to demonstrate the physical effect of the hot carrier fast lane, and to see if it can enhance THz generation at a fixed voltage. This has been demonstrated clearly, and is the focus of this manuscript.

Second, our goal was not to maximize the far-field emission power. In that case, indeed we would want to operate the devices at their maximum bias voltage, and compare maximum power output. This would indeed be interesting to do in a future study (hopefully this work will stimulate such work). For the study reported here, however, we did not push the maximum voltage of the graphene-enhanced switch, since we did not focus on optimizing the process and packaging required to make that a meaningful measurement; we confine our attention to comparisons at fixed voltages to elucidate the effect.

Most importantly, we are working in a transmission line geometry on silicon. For on-chip applications of THz, especially on Si, the relevant bias voltages will be in the range of 5V or so, which our devices can handle. Many on-chip applications of THz electronics will not have voltages in the 20-V range or so on chip. Hence we believe it is sufficient for this paper to report the THz enhancement due to the physical effect under consideration, and leave the maximization of power emission to some future study. Indeed, a little further later in his report, this referee suggest we split the paper into two for reasons of scope. We disagree with that recommendation, and believe that the well-defined scope of this study stands by itself as a coherent, self-contained contribution.

Comment 6

Line 115-118: It's not clear how showing strongest THz generation when excitation occurs near the electrode eliminates the possibility of edge effects dominating. If I'm following, wouldn't higher emission when the edge is illuminated with the pump support the possibility of edge effects? Please elaborate on these claims.

We appreciate the referee's thoughtful comments. We revised the discussion on possibility of edge effects. We do want to emphasize on the following:

The PTE effect and built-in field induced photovoltaic effects shall have much sharper position dependence than our observation, given the very small beam diameter ($< 2 \mu\text{m}$) and long channel ($15 \mu\text{m}$). Nevertheless, the linear channel-bias dependence and negligible emission under zero bias suggests that edge effects are not dominating the THz field generation. We recognize that such spatial dependence may not be the best evidence we have, even though it can be well-explained. Hence, we moved the related discussion to the supplementary and discussed the possibilities with the channel bias dependence in the revised manuscript.

Comment 7

Lines 119-121: Please concisely state up front what is explicitly being tested by studying these two special cases of the device. My understanding is that the motivation for the graphene only device is to remove any photoconductive effect from the substrate, but I'm not understanding how the examination of the low O implanted device supports the claim that THz emission is dominated by absorption in the Si layer.

We thank the referee for the comment. As stated in the manuscript, the two control groups are designed to verify that the THz generation is a direct result of pump-beam absorption in the silicon layer, rather than any effect intrinsic to graphene.

Specifically, we test the low O+ flux sample to verify that the generated THz signal's bandwidth is determined by the carrier lifetime in silicon rather than in graphene. If the THz generation is caused by some Si-graphene interactions that does not involve silicon absorption (for example, doping, surface passivation, or dielectric environment), it is unlikely to be so strongly dependent on silicon carrier lifetime.

Comment 8

Lines 155-156: The claim "this is a clear sign of..." is not clear to me at all. I'm just not following the logical flow from the rest of the paragraph, and I'm not sure what this claim is trying to state. Please revise and clarify.

We revised the paragraph to better interpret the underlying physics from experimental observation:

“We further probed the photocurrent of the device at zero channel bias ($U=0$) and sweeping gate bias in a spatially resolved manner (Fig. 4b). Specifically, we examine the photocurrent at regions sufficiently far from the metal edges. The photocurrent at such regions cannot come from horizontal electric fields, since neither the built-in field from contacts nor any channel bias exists. A vertical photocurrent from silicon to graphene, followed by excess carrier diffusion to the edge, however, can contribute to the photoresponse. We observe a sign flip of photocurrent when sweeping V_g from negative to positive (pink dotted line in Fig. 4b). The flipped direction of photocurrent is only possible if V_g inverses the direction of the built-in field at the graphene-silicon interface.”

Meanwhile, the paragraph next to it works as a further explanation.

Comment 9

In general, the final 8 paragraphs of the results section (lines 112-202) are unclear and difficult to follow. It's clear the authors performed a lot of work, but I think it is trying to be condensed to fit all the results into a single paper.

I would highly recommend the authors consider break this work into two papers, one focusing on characterizing the extent of the enhancement itself, addressing some of the comments I've made here, and another paper more fully investigating the mechanisms of the enhancement. There seems to be a lot of good work here that could be expanded into two high quality papers, but as it stands now there is a lot of unclear and difficult to follow logic behind the conclusions that are made.

We thank the referee for the valuable comment. Perhaps we included too many things in the discussion, which digressed from the main point and led to confusion. To address this, we revised the text accordingly and:

1. Removed the spatial dependence to the supplementary. As also pointed out by the referee in another comment, the discussion does not work as the best evidence we have for ruling out the possible competing effects. We keep it in the supplementary for documentation.
2. Moved the extended discussions on pure graphene devices to supplementary. The supplementary discussions only function as a comparison to other previously reported THz generation in similar pure graphene devices. We agree that it is better to use the supplementary, rather than the main manuscript, to explain why the device behaves very differently from other reported works.

3. Revised a few other sections to make the discussion clearer and easier to follow.

Minor Comments

Line 57: Fig 1 is, in general, not very clear. Some additional text/labels would be useful to clarify what each part of each image is. Additionally, being that Fig 1a, Fig 1b, and Fig1c are all referenced in different sections of the paper (1a in the Intro, 1b in the Fab and Characterization, 1c in the results) it might flow better breaking the figure up into three different figures.

We revised the figures and captions to fix this.

Line 41 and 78: The claim of nanopatterning to achieve nanoscale plasmonic electrodes is valid, but to someone such as myself who is unfamiliar with graphene processing, wet-transfer of a graphene film also sounds difficult. Is this a more (or less) scalable process than nanolithography? Please provide a reference supporting that it is, or reconsider the claim that there are processing advantages to this type of device over current nanopatterned electrode configurations.

We thank the referee for this question. We used the wet transfer to process graphene since it is the most convenient method in our lab. Nowadays there are mature, wafer scale roll-to-roll transfer methods that are compatible with mass production in the semiconductor industry. We added the following sentences to the graphene device fabrication paragraph:

“Notably, adding graphene does not necessarily much increase the fabrication complexity, as mature wafer-scale graphene growth, transfer, and device fabrication have been reported.” (Reference added)

Fig 2 and 3: Please add legends or some other visual indicator to each graph to help clarify what each trace represents. For example, 2a, where there are many curves, something like an upward arrow through the curves and text stating “increasing channel bias” would help.

Done.

Fig 4c is not clear, without extensively reading the text and filling in gaps as to what represents what in the illustration. Please add clarification to help the image become more understandable from a “stand-alone” perspective.

We thank the referee again for the advice and revised Fig. 4, together with its discussions.

Reviewer #1 (Remarks to the Author):

The authors have addressed my comments well and I would now recommend publication of the manuscript.

Reviewer #2 (Remarks to the Author):

1. From a general perspective the main problem I found is that there is no clear demonstration of the capacity of this device to produce a reasonable amount of THz power that can be coupled out of the device. In this sense, there are two aspects that concern me. Firstly, the study does not demonstrate actual emission of terahertz radiation in the far field, which would be the real motivation for it. In other words, the emitter and detector lie on the same substrate within 100 μ m, which is a fraction of a wavelength, from each other, therefore this does not show that actual radiation is coupled out of the device, and the effect itself might be only a capacitive effect in this small device.

We appreciate the referee's comment and correspondingly have changed "terahertz emission" to "terahertz generation" in the title and the rest of the manuscript to better reflect that far-field radiation is not reported in this work.

We do not agree, however, that only far-field radiation is interesting or important for THz generation. There are many applications of THz fields confined to waveguides or transmission lines. While there are some important differences between radiation from a photoconductive switch into a guided mode and to free space from an antenna (which will result in different pulse shapes, bandwidth, etc.), the fundamental process of generating a transient photocurrent is essentially the same in both cases. The scope of our research program reported here was confined to transmission line experiments; we would enthusiastically hope that other groups (or our own, subject to future funding support) might be able to further develop the concept we present here for free space applications, but we believe the concept and physics presented here in the context of transmission line applications is fully sufficient for publication.

Finally, we do note that we performed control experiments on Si with the same geometry but without graphene, as well as control experiments done on graphene (without Si), which rule out a capacitive effect.

I thank the authors for their response. Yet I do not completely agree. The demonstration of a signal that is produced and detected within a small fraction of a wavelength is of very limited relevance and in my opinion is not even a clear demonstration of higher power being generated. With this I do not want to say that your claims are false, just that they are not strongly supported. For example, if you think of a waveguide where a part of it is coated with a conductive layer (graphene for example) it is clear that the transmission will be affected, this does not mean that more/less power was delivered at the input of the waveguide, only that it will propagate differently, and that is why I insist that far field measurements are much more conclusive in terms of demonstrating higher powers generated.

Secondly, there is no clear indication that the device can sustain a large enough bias voltage to produce a significant emission. A dark resistivity measurement is not presented which would demonstrate that for instance \sim 10-25V across a 10-20 μ m gap can be tolerated by the device, which are typical values for THz emitters currently used, with just 3V or so, the amount of power emitted will actually be lower than that of a Si-GaAs emitter with a bias of 10 to 25V. A comparative measurement with a standard emitter would be desirable in order to be able to make claims about the actual improvement.

Two points should be noted:

(1) We would like to point out that the devices were biased up to 9V instead of the 3V quoted, as shown in Fig.2 and Fig. 3. Because the devices were not passivated and without optimized packaging, and all tests were carried out in air under ambient condition, we intentionally avoided applying a high voltage and did not perform damage testing; this would remain for future work.

(2) All control experiments were conducted under the same biasing and testing conditions; hence we focused on comparing Si-based devices with and without the graphene in direct comparison in order to understand the relevant device physics. Optimization of power (including maximizing the applied voltage) remains for future work; the physical conclusion regarding enhancement at a given bias is robust. (Note: referee #3 comment #5 raises a similar point; please see our additional discussion there.)

I understand that the authors did not want to test their devices to the break-down point, however, this is an important issue. If the actual voltage sustained by the device under real operation conditions is significantly lower than other more conventional devices their practical viability might be questionable, therefore even if this is the case, the issue should be acknowledged and discussed.

2. While very short carrier lifetime is indeed required for good photoconductive detectors, this is in general not true for photoconductive emitters. For emitters the contrast between dark and bright conductivity and the possibility of high bias voltage are the main characteristic that determines its output power. This is wrong all along the narrative in the current manuscript. The "short" carrier lifetime is needed in the case of silicon because the lifetime in non-damaged-silicon is so long that the conductivity remains almost unchanged for hundreds of nanoseconds, which means that the repetition period of the laser (in the several tens of MHz) is shorter than the lifetime, and the silicon remains conductive for the entire duration of the laser repetition cycle (~10ns).

The referee is correct that, for free-space THz emission, it is possible to use longer-lifetime semiconductors, relying on the fact that the far-field emission is proportional to dJ/dt . Sometimes semi-insulating GaAs is used, for example, although LT-GaAs is also frequently used, where the growth parameters are chosen such that picosecond lifetimes are employed. Applications using Si generally employ damage producing ps lifetimes. Since many THz implementations do employ short-lifetime material, the effect discussed in our work (the "hot carrier fastlane") will be directly relevant to those implementations.

The most important point, however, is that for THz pulses on transmission lines, which is what we are concerned with in this study, ps lifetimes are required. Otherwise the voltage waveform is a step function, which is not useful for THz modulation, studies of device transient phenomena, or almost any other application employing confined fields in waveguides/transmission lines. Hence we focus exclusively on ps-lifetime material in this study.

We note that in the revised manuscript, we have made clear that we are not reporting free-space THz emission. We have also revised the abstract and introduction to confine our discussion of the preference of short carrier lifetime to PCS coupled with transmission lines/waveguides.

I thank the authors once more. However, I would like to point out that their argument is wrong. The current, even in a long-carrier-lifetime substrate, is not a step-function since the transient carrier separation produces a dipole pointing in the direction opposing the external electric field, which in turn stops the current, this is why SI-GaAs emitters work well and produce bipolar pulses.

Furthermore, the sentence that now appears in lines 34-37 completely lacks justification, please either provide a reference that states exactly that, a good physical argumentation or remove it.

3. The power comparison of the graphene vs the non-graphene samples is more or less fair, this is not the case when comparing the devices with different implantation states, since the implantation will heavily influence the performance of the detector, which in this case is on the same substrate, those comparisons are therefore inappropriate/unfair.

We would like to point out that the purpose of the control on Si with different implantation is to confirm that the absorption mainly happens in the silicon and not the graphene; the Si switch performance is carefully made identical to the other devices for this particular control by running a second implantation only to the detector region. We further clarify the issue with the following sentences added to the control experiment section:

"Additionally, we performed a second O⁺ ion implantation to the detector region only. The total implantation flux in the region is identical to the heavily implanted samples, so that the detector's temporal resolution and sensitivity are identical to those used in other samples."

Thanks for clarifying, this is more reasonable. Yet I am not sure if an implantation by steps is equivalent.

4. The argument presented between lines 101 and 111 is hard to follow. the RCL model is referred to without any explanation of what RCL circuit was assumed, and how that is linked to the actual device presented here.

We thank the reviewer for the valuable advice. The key point here is that we observe a strong (25-time) enhancement of SNR, despite a larger noise due to the fact that graphene layer lowers the channel resistance. A lower channel resistance can lead to larger thermal noise, which is supported by the cited RCL modeling reference. However, we did not perform RCL modellings in this work.

We have revised the manuscript to avoid any potential confusion.

Thanks!

**5. Most of the discussion presented between line 112 and 130 does not make much sense. Furthermore, the higher emission when exciting a photoconductive emitter near the anode is very well documented experimentally [Appl. Phys. Lett. 59,1972 (1991); IEEE J. Quantum Electron. 32, 1664 (1996)], and explained theoretically [Phys. Rev. B 71, 195301 (2005)].
1972 (1991)**

We appreciate the referee's comments. We have revised the paragraph, moved the less straightforward discussions on spatial resolution to the supplementary material, discussed the possibility of edge effects with other stronger evidence, and cited the above references. We do want to emphasize the following:

1. The purpose of this portion of the discussion is to understand the mechanism for the observed strong enhancement between device with and without graphene layer, not the enhancement effect when near anode as has been studied previously in the quoted reference.

2. The PTE effect and built-in field induced photovoltaic effects will have much sharper position dependence than our observation, given the very small beam diameter (< 2 μm) and long channel (15 μm). Nevertheless, the linear channel-bias dependence and negligible emission under zero bias suggests that edge effects are not dominating the THz field generation.

3. A minor point, the stronger emission near anode is attributed to an uneven field distribution at the reverse-biased contact in the quoted literature. It may be very different given that the graphene/metal junction has a much weaker built-in field (graphene is a semi-metal). Further study on this might be needed, but is outside of the scope of current paper.

Thanks!

4. The authors claim a lifetime of their substrates of 0.7ps (line 78). How did they determine this? This is actually critical for the performance of the detector.

We followed the existing literature, specifically references 28 and 29, to estimate the lifetime from the transient measurement. We used the standard and well-accepted procedure and dose. Beyond a threshold dose the lifetime does not further decrease below 0.7 ps, which is mentioned in the reference. Supplementary Fig. 2c shows the pulse generated by a simple silicon emitter. The observed pulse width is actually longer than the actual detector ON time, considering the dispersion of the THz field during its propagation along the waveguide.

What do you mean "from the transient measurements"? I am afraid that although ion implantation produces far more reproducible samples than low-temperature growth, small changes in the temperature, time or even the atmosphere of the annealing can lead to very different lifetimes, so a better determination, such as optical-pump terahertz-probe of the carrier lifetime in this case is very important in order to have a convincing argument. Else the authors should acknowledge that this is only a rough estimation.

5. The size of the gap between contacts is never described in the manuscript, and it is a very important parameter, I estimated it to be of the order of 100um by looking at the schematics and the non-scaled photograph in

We thank the referee's comment. The gap size is 15 um. We apologize for any confusion, and have now clearly stated the device geometry in the device fabrication portion.

Thanks

6. The discussion presented between lines 147 and 166 was extremely confusing to me. The authors refer to "the gate" without defining it, then the voltage V_g is mentioned, but it is not clear if this is the same voltage that is referred to as U in Fig.1. The fact that there is a current at 0-voltage (shown in Fig.4a) does not make much sense. All-in-all I believe this paragraph was written without much care, and is confusing, I must confess that after reading it 3 times I could not make any sense out of it.

We thank the referee's comments and apologize for any confusion. Even though all device structure and biasing info, including bias voltage, gate voltage, were defined either in the main text or the supplementary information, it can be confusing for someone not familiar with a graphene device. We have made effort to clarify that in the revision with both texts and figures.

For the specific questions mentioned above: (1) the gate is referred to the gate electrode fabricated on top of the hybrid PCS structure (see supplementary Section I). We also added a schematic of the gate to Fig. 4a inset in the main manuscript. (2) 0-voltage in Figure 4a means $V_g = 0$ V on the gate, not the bias voltage. The x-axis for Fig. 4a is gate voltage.

I respectfully point out to the authors that there is no need to take the comments personal or reply aggressively, my comments were not intended as an aggression. I consider myself a person that is very familiar with terahertz devices, having worked in the field for almost two decades. With all due respect a clear explanation of how you call your electrodes, perhaps with a clear diagram is something that I consider a reasonable request. Furthermore, in the terahertz community the typical nomenclature for a photoconductive switch is cathode and anode, which are typically the only two electrodes present, since it is a switch, not a mosfet with a gate and a channel. Even if you opt to keep using the FET nomenclature for the electrodes, a clear explanation will make it easy to understand for everyone what is meant. Perhaps names in Fig 1 would

get rid of all ambiguities.

7. Between lines 167 and 169, a description of what is seen in Fig 4d is presented, but this does not match what is seen in the figure. For instance the authors discuss voltages between -3V and +7V, but in the figure only positive voltages are shown. Furthermore, the minimum of the "M"-curve appears at 3V for one of the two curves, but not for the other. All in all this part was also confusing, and there are inconsistencies between the text and the plot.

We appreciate the referee's comment and have revised both the text and the figure to clarify the observed trend.

"Fig. 4d shows the field's amplitude captured by the X and Y channels of a lock-in amplifier, which reflects both the absolute value and the phase of the signal. The amplitude's absolute values peak at around -3 V and 7 V (orange dotted lines), but dips around 3 V (green dotted line), leading to a pair of M-shaped dependence (the red M positioned upward, and the blue M mirrored by the horizontal axis due to the phase of Y channel). "

Thanks

8. The paragraph between lines 183 and 190, also make claims comparing the results in figure 2 and 3, however, for the reasons already mentioned, about the detector being strongly affected by the carrier lifetime, it is not possible to make fair comparisons.

As we explain in previous response, the detectors in two devices are implanted with the same dose while the PCS channel is doped differently. Hence the comparison is fair. We hope the revised description is clear to our readers.

Thanks

9. Other specific comments:

a. line 12: "To obtain picosecond ultrafast pulses, semiconductors are typically heavily damaged to reduce the carrier lifetime". Please reconsider

Unfortunately, the reviewer's comment here is incomplete, probably by mistake, so we are not quite sure how to address it. In response, to this comment, we revised the text from 'heavily damaged' to 'with high defect density'.

I believe "heavily damaged" is not a defined term, I understand that you mean ion-implantation, but "heavy damage" could be achieved by dancing on the sample for 20 minutes, or by throwing it from the third floor of a building. Yet, even if we refer to damage by ion implantation the "heavy" part is completely subjective, is a dose of 1 or 10 or 10^{20} ions/cm² heavy or where do you draw the line?

b. line 28: "channel", please consider using the more common term "gap"

Thanks for the comment. We feel that using 'gap' and 'channel' has different benefits throughout the paper. The word 'gap' is indeed widely used by convention in many PCS literatures, while our choice of channel better describes our device following the convention of transistors. To avoid confusion, we added the following words to the manuscript:

"(or a gap, as it is often referred to in the literature)"

Ok

c. line 30: idem

Idem means "the same as the previous comment"

d. lines 30-32: please reconsider the sentence starting "The transient current ..."

This is not a clear comment/critique, so we are not quite sure how to address it.

"The transient current in *a* PCS couples with a*n* antenna to emit the THz *electromagnetic radiation* to the far field"

e. line 33: the authors talk about "low quantum efficiency", yet this is not a term that is normally used at least in the THz community, it would be easy to find efficiencies that are not that low since THz photons contain about 3 to 4 orders of magnitude less energy than IR-VIS photons, one could have "quantum efficiencies" close to 1 and still convert 0.1% to 0.01% of the laser energy into terahertz energy. Perhaps "low conversion efficiency" is more appropriate, although in general this is also not a parameter that is typically reported, as the bias voltage also plays a fundamental role in the conversion, so it is not a property of the device as such.

We agree with the referee that "conversion efficiency" is a more relevant term, and have modified the text accordingly.

Thanks

f. line 35: "carrier recombination", I believe the authors refer to carrier trapping, since defects act as traps, but this is a different process to the recombination.

We changed 'carrier recombination' to 'carrier lifetime'.

Thanks

g. line 36-37: the carrier mobility drops, but not by "a few orders of magnitude" please reconsider

We changed it to 'drops substantially'.

Thanks

h. line 42: "reducing the overall device active area" This is actually not entirely true, the patterned antennas actually allow larger laser beams, which results in a larger emitter area, in the case of plasmonic contacts, they act a bit like "light funnels" so in the end the effective area is not necessarily reduced.

We removed the words to avoid any controversy.

Thanks

i. line 73. Although "pump-probe" has been wrongly used by many authors, in general the THz community understands as a pump-probe setup the one where a sample (not the emitter or detector) is optically pumped by a third laser pulse, and probed by the THz pulse. The more accepted, and less ambiguous, term used now is THz-generation (beam) and gate (detector beam).

We understand there are different terminology regarding this within the literature. We do believe our terminology used in this work is fully within the conventions of much of the literature, and is completely unambiguous in the context of our text. In order to remove any possible ambiguity, we have added "or gate" to the text (line 95) to indicate that the probe pulse indeed gates the signal.

Thanks, perhaps names in Fig 1 would get rid of all ambiguities

j. The dimensions of the device are not mentioned in the manuscript, please include them

Done

Thanks

k. The repetition rate of the laser and power of each beam are also not mentioned in the device fabrication and characterization section

We've added the repetition rate (76 MHz) in the revised manuscript. The pump and probe power are described in the captions of the relevant figures.

l. Your pulses look indeed Gaussian, but the typical shape of THz waveform pulses is bipolar, or even more complicated, the pulses should integrate zero over time, this is for charge conservation reasons. Perhaps your current signals require further processing, I believe this is related to the carrier lifetime not being short enough in the detector.

The referee is correct in his/her observation that the waveform is not bipolar. In fact, it is not required that the pulse be bipolar with an obvious zero time integral within the experimental scan range. It is a common misconception that this is required, but in fact this is only required in the FAR FIELD of a source. (Of course, the integral must be rigorously zero in the far field, but it can indeed be nonzero near the emitter). This subject has been dealt with in great detail in a classic paper by Phil Bucksbaum and his student D. You back in the 1990's; the reference is:

D. You and P. H. Bucksbaum, "Propagation of half-cycle far infrared pulses," J. Opt. Soc. Am. B 14, 1651-1655 (1997).

That paper shows that THz pulses can appear as quasi-unipolar for quite surprising distances, and this is the case with our experimental pulses. The detection is not fully in the far field, as the signal is confined to a transmission line near the source.

Now, it is true that there must be a negative component that would eventually integrate to zero. However, the amplitude of this component can be so small that it is completely within the noise limit of the data, and correspondingly the negative "tail" can be as long as nanoseconds (corresponding essentially to the re-charging of the semiconductor gap).

We have discussed the issue and added this reference in the supplementary, so that any reader may find this, but a long discussion of the pulse shape would not be necessary, as the above reference is sufficient.

Thanks, I must point out that this does not match my experience.

m. line 128: there si an error (I think a reference)

Corrected.

Thanks

n. line 179: idem.

Corrected.

Thanks

Point-by-point Responses to Reviewers' Comments

Reviewers' comments are in **black**; our new responses are in **blue**. Our previous response is indicated in **red** to avoid confusion.

Reviewer #1 (Remarks to the Author):

The authors have addressed my comments well and I would now recommend publication of the manuscript.

Thanks.

Reviewer #2 (Remarks to the Author):

1. From a general perspective the main problem I found is that there is no clear demonstration of the capacity of this device to produce a reasonable amount of THz power that can be coupled out of the device. In this sense, there are two aspects that concern me. Firstly, the study does not demonstrate actual emission of terahertz radiation in the far field, which would be the real motivation for it. In other words, the emitter and detector lie on the same substrate within 100um, which is a fraction of a wavelength, from each other, therefore this does not show that actual radiation is coupled out of the device, and the effect itself might be only a capacitive effect in this small device.

We appreciate the referee's comment and correspondingly have changed "terahertz emission" to "terahertz generation" in the title and the rest of the manuscript to better reflect that far-field radiation is not reported in this work.

We do not agree, however, that only far-field radiation is interesting or important for THz generation. There are many applications of THz fields confined to waveguides or transmission lines. While there are some important differences between radiation from a photoconductive switch into a guided mode and to free space from an antenna (which will result in different pulse shapes, bandwidth, etc.), the fundamental process of generating a transient photocurrent is

essentially the same in both cases. The scope of our research program reported here was confined to transmission line experiments; we would enthusiastically hope that other groups (or our own, subject to future funding support) might be able to further develop the concept we present here for free space applications, but we believe the concept and physics presented here in the context of transmission line applications is fully sufficient for publication.

Finally, we do note that we performed control experiments on Si with the same geometry but without graphene, as well as control experiments done on graphene (without Si), which rule out a capacitive effect.

I thank the authors for their response. Yet I do not completely agree. The demonstration of a signal that is produced and detected within a small fraction of a wavelength is of very limited relevance and in my opinion is not even a clear demonstration of higher power being generated. With this I do not want to say that your claims are false, just that they are not strongly supported. For example, if you think of a waveguide where a part of it is coated with a conductive layer (graphene for example) it is clear that the transmission will be affected, this does not mean that more/less power was delivered at the input of the waveguide, only that it will propagate differently, and that is why I insist that far field measurements are much more conclusive in terms of demonstrating higher powers generated.

On this point we will have to respectfully disagree with the referee, and leave it to the editor's judgment. The reason is that, while we perfectly agree that free-space THz applications are extremely important and ever-growing, we do believe there are critical applications in waveguide (essentially, on-chip) geometries. Ultrafast measurements on nanoelectronic devices, ballistic conductors, high-speed on-chip switching devices, etc. are often best performed on-chip. Indeed, it is to be expected that the continued development of high-speed nanoelectronic and nano-optoelectronic devices will be closely tied to THz generation and measurements on-chip. Not everything is free-space.

Now, we do agree with the reviewer that experimental proof of higher-efficiency free-space THz generation would be most desirable. However, we are presently not in a position to perform such

a set of measurements in a timely fashion, and this will have to wait for further development. We believe that the physical principle developed in this work does stand alone.

We have modified the conclusions statement slightly to reflect this point of view. Specifically, we added the following sentence in the conclusion remark: “The most direct implications of our findings are for THz generation enhancement in waveguide (essentially, on-chip) geometries. Ultrafast measurements on nanoelectronic devices, ballistic conductors, high-speed on-chip switching devices, etc. are often best performed on-chip. Our results also point toward likely enhancement of free-space THz generation by engineering the hot-carrier fast lane, but experimental proof of free-space THz emission will be needed in further studies.”

Secondly, there is no clear indication that the device can sustain a large enough bias voltage to produce a significant emission. A dark resistivity measurement is not presented which would demonstrate that for instance $\sim 10\text{-}25\text{V}$ across a $10\text{-}20\mu\text{m}$ gap can be tolerated by the device, which are typical values for THz emitters currently used, with just 3V or so, the amount of power emitted will actually be lower than that of a Si-GaAs emitter with a bias of 10 to 25V . A comparative measurement with a standard emitter would be desirable in order to be able to make claims about the actual improvement.

Two points should be noted:

(1) We would like to point out that the devices were biased up to 9V instead of the 3V quoted, as shown in Fig.2 and Fig. 3. Because the devices were not passivated and without optimized packaging, and all tests were carried out in air under ambient condition, we intentionally avoided applying a high voltage and did not perform damage testing; this would remain for future work.

(2) All control experiments were conducted under the same biasing and testing conditions; hence we focused on comparing Si-based devices with and without the graphene in direct comparison in order to

understand the relevant device physics. Optimization of power (including maximizing the applied voltage) remains for future work; the physical conclusion regarding enhancement at a given bias

is robust. (Note: referee #3 comment #5 raises a similar point; please see our additional discussion there.)

I understand that the authors did not want to test their devices to the break-down point, however, this is an important issue. If the actual voltage sustained by the device under real operation conditions is significantly lower than other more conventional devices their practical viability might be questionable, therefore even if this is the case, the issue should be acknowledged and discussed.

We respectfully disagree with the referee's assessment. As we explain in previous response, our devices were tested up to 9V, rather than only at 3V, and was in line with typical biasing voltages for a silicon-on-sapphire photoconductive switch in literature. At higher voltage, un-passivated graphene devices in general risk breakdown due to oxidation. We agree that it is technically important to develop proper passivation/packaging for our devices and push it to even higher voltage, but it does not change the physics of the enhancement effect we observed here, and should be discussed in a future work.

2. While very short carrier lifetime is indeed required for good photoconductive detectors, this is in general not true for photoconductive emitters. For emitters the contrast between dark and bright conductivity and the possibility of high bias voltage are the main characteristic that determines its output power. This is wrong all along the narrative in the current manuscript. The "short" carrier lifetime is needed in the case of silicon because the lifetime in non-damaged-silicon is so long that the conductivity remains almost unchanged for hundreds of nanoseconds, which means that the repetition period of the laser (in the several tens of MHz) is shorter than the lifetime, and the silicon remains conductive for the entire duration of the laser repetition cycle (~10ns).

The referee is correct that, for free-space THz emission, it is possible to use longer-lifetime semiconductors, relying on the fact that the far-field emission is proportional to dJ/dt . Sometimes semi-insulating GaAs is used, for example, although LT-GaAs is also frequently used, where the growth parameters are chosen such that picosecond lifetimes are employed. Applications using Si

generally employ damage producing ps lifetimes. Since many THz implementations do employ short-lifetime material, the effect discussed in our work (the “hot carrier fastlane”) will be directly relevant to those implementations.

The most important point, however, is that for THz pulses on transmission lines, which is what we are concerned with in this study, ps lifetimes are required. Otherwise the voltage waveform is a step function, which is not useful for THz modulation, studies of device transient phenomena, or almost any other application employing confined fields in waveguides/transmission lines. Hence we focus exclusively on ps-lifetime material in this study.

We note that in the revised manuscript, we have made clear that we are not reporting free-space THz emission. We have also revised the abstract and introduction to confine our discussion of the preference of short carrier lifetime to PCS coupled with transmission lines/waveguides.

I thank the authors once more. However, I would like to point out that their argument is wrong. The current, even in a long-carrier-lifetime substrate, is not a step-function since the transient carrier separation produces a dipole pointing in the direction opposing the external electric field, which in turn stops the current, this is why SI-GaAs emitters work well and produce bipolar pulses. Furthermore, the sentence that now appears in lines 34-37 completely lacks justification, please either provide a reference that states exactly that, a good physical argumentation or remove it.

The referee is indeed correct that the current can be a pulse and not a step function; our argument was not sufficiently clear. The behavior depends on the contacts as well as the coupling to the radiation field (for free-space emission), which is a subject one of the authors investigated in some detail in the past (ref: W. Sha, J. Rhee, T.B. Norris, and W.J. Schaff, "Transient Carrier and Field Dynamics in Quantum Well Parallel Transport: from the Ballistic to the Quasi-equilibrium Regime," IEEE J. Quant. Electron. 28, (special issue on Ultrafast Optics and Electronics), 2445 (1992).

If the contacts between the electrodes and the semiconductor are non-Ohmic, then indeed the movement of space charge in the gap due to an applied field will act to screen the bias field, shutting off the current (measurements of the screening dynamics within the gap were performed

in the above reference). The result will be a voltage pulse applied to a transmission line, or a bipolar pulse radiated into free space. So, we agree with the reviewer that a free-space bipolar pulse can be generated in long-lifetime semiconductors.

However, if the contacts are ohmic, then carriers are collected at the electrodes and current continues to flow down the transmission line. Space charge does not build up. Once a semiconductor becomes conducting due to photoinjection, and the contacts are ohmic, the gap region continues to be conducting until recombination (or trapping) occurs, and a voltage step function is obtained. The classic original measurement of this may be found in figure 4 of J.A. Valdmanis and G.A. Mourou, IEEE J. Quant. Electron. QE-22, 69 (1986), which we reproduce here for the convenience of the reader:

the trans-
odulator.
trodeless
ion we
each.
a con-
e elec-
5). Ini-
n with
n thick
c axis,
. Elec-
crystal
g beam
crystal
ngle of
rad in

lithium tantalate crystal. The detector essentially generates a step function on the picosecond time scale when it is triggered by subpicosecond optical pulses. The sampled rise time is a direct measure of the temporal resolution of the sampling system [14]. Fig. 4 shows the detector and crystal in the velocity matched geometry and the accompanying measured subpicosecond rise time. The foot on the leading edge of the curve is an indication that dispersive effects are already present after less than 0.2 mm of propagation.

B. Transverse Coplanar Line Modulator, Fig. 3(b)

An alternate method of fabricating traveling wave modulators that avoids the difficulty of a balanced line on a thin substrate is the transverse field, coplanar line modulator [27] as depicted in Fig. 3(b). Coplanar waveguide and coplanar strip transmission lines have been used frequently in microwave integrated circuit applications. They

Regarding the sentence in line 34-37: We thank the referee's comment and apologize for any ambiguity. We agree that longer carrier lifetime semiconductors (with high carrier mobility), such as semi-insulating GaAs, can be designed into efficient broadband THz emitter by incorporating clever designs, such as the adoption of interdigitated antenna and the use of high-power, low repeat optical pump. Our statement was related to PCS coupled to waveguide (on-chip) scenario, which was our attempt to limit the discussion based on previous review comments. To make it

clear, we further modify the sentence as following and include a reference for work on semi-insulating GaAs:

“While substrates with longer carrier lifetime (such as semi-insulating GaAs) can be used for efficient broadband THz emitter though clever engineering (such as the use of interdigitated antenna)⁷, short carrier lifetimes (picosecond) are in general preferred for efficient THz generation...”

3. The power comparison of the graphene vs the non-graphene samples is more or less fair, this is not the case when comparing the devices with different implantation states, since the implantation will heavily influence the performance of the detector, which in this case is on the same substrate, those comparisons are therefore inappropriate/unfair.

We would like to point out that the purpose of the control on Si with different implantation is to confirm that the absorption mainly happens in the silicon and not the graphene; the Si switch performance is carefully made identical to the other devices for this particular control by running a second implantation only to the detector region. We further clarify the issue with the following sentences added to the control experiment section:

“Additionally, we performed a second O⁺ ion implantation to the detector region only. The total implantation flux in the region is identical to the heavily implanted samples, so that the detector’s temporal resolution and sensitivity are identical to those used in other samples.”

Thanks for clarifying, this is more reasonable. Yet I am not sure if an implantation by steps is equivalent.

4. The argument presented between lines 101 and 111 is hard to follow. the RCL model is referred to without any explanation of what RCL circuit was assumed, and how that is linked to the actual device presented here.

We thank the reviewer for the valuable advice. The key point here is that we observe a strong (25-time) enhancement of SNR, despite a larger noise due to the fact that graphene layer lowers the channel resistance. A lower channel resistance can lead to larger thermal noise, which is supported by the cited RCL modeling reference. However, we did not perform RCL modellings in this work. We have revised the manuscript to avoid any potential confusion.

Thanks!

5. Most of the discussion presented between line 112 and 130 does not make much sense. Furthermore, the higher emission when exciting a photoconductive emitter near the anode is very well documented experimentally [Appl. Phys. Lett. 59,1972 (1991); IEEE J. Quantum Electron. 32, 1664 (1996)], and explained theoretically [Phys. Rev. B 71, 195301 (2005)].

1972 (1991)

We appreciate the referee's comments. We have revised the paragraph, moved the less straightforward discussions on spatial resolution to the supplementary material, discussed the possibility of edge effects with other stronger evidence, and cited the above references. We do want to emphasize the following:

1. The purpose of this portion of the discussion is to understand the mechanism for the observed strong enhancement between device with and without graphene layer, not the enhancement effect when near anode as has been studied previously in the quoted reference.

2. The PTE effect and built-in field induced photovoltaic effects will have much sharper position dependence than our observation, given the very small beam diameter ($< 2 \text{ m}$) and long channel (15 m). Nevertheless, the linear channel-bias dependence and negligible emission under zero bias suggests that edge effects are not dominating the THz field generation.

3. A minor point, the stronger emission near anode is attributed to an uneven field distribution at the reverse-biased contact in the quoted literature. It may be very different given that the

graphene/metal junction has a much weaker built-in field (graphene is a semi-metal). Further study on this might be needed, but is outside of the scope of current paper.

Thanks!

4. The authors claim a lifetime of their substrates of 0.7ps (line 78). How did they determine this? This is actually critical for the performance of the detector.

We followed the existing literature, specifically references 28 and 29, to estimate the lifetime from the transient measurement. We used the standard and well-accepted procedure and dose. Beyond a threshold dose the lifetime does not further decrease below 0.7 ps, which is mentioned in the reference. Supplementary Fig. 2c shows the pulse generated by a simple silicon emitter. The observed pulse width is actually longer than the actual detector ON time, considering the dispersion of the THz field during its propagation along the waveguide.

What do you mean “from the transient measurements”? I am afraid that although ion implantation produces far more reproducible samples than low-temperature growth, small changes in the temperature, time or even the atmosphere of the annealing can lead to very different lifetimes, so a better determination, such as optical-pump terahertz-probe of the carrier lifetime in this case is very important in order to have a convincing argument. Else the authors should acknowledge that this is only a rough estimation.

Here we follow one of the common techniques used in the literature (such as ref 28, 29) for the lifetime estimate. “from the transient measurements” refers to the time-delay measurements from which the references obtain their lifetime estimate. We don’t disagree that optical-pump terahertz-probe of carrier lifetime is likely more accurate. In fact, in our own data (Fig. 2c) and discussion, we simply refer them to rise time and fall time.

5. The size of the gap between contacts is never described in the manuscript, and it is a very important parameter, I estimated it to be of the order of 100um by looking at the schematics and the non-scaled photograph in

We thank the referee's comment. The gap size is 15 um. We apologize for any confusion, and have now clearly stated the device geometry in the device fabrication portion.

Thanks

6. The discussion presented between lines 147 and 166 was extremely confusing to me. The authors refer to "the gate" without defining it, then the voltage V_g is mentioned, but it is not clear if this is the same voltage that is referred to as U in Fig.1. The fact that there is a current at 0-voltage (shown in Fig.4a) does not make much sense. All-in-all I believe this paragraph was written without much care, and is confusing, I must confess that after reading it 3 times I could not make any sense out of it.

We thank the referee's comments and apologize for any confusion. Even though all device structure and biasing info, including bias voltage, gate voltage, were defined either in the main text or the supplementary information, it can be confusing for someone not familiar with a graphene device. We have made effort to clarify that in the revision with both texts and figures.

For the specific questions mentioned above: (1) the gate is referred to the gate electrode fabricated on top of the hybrid PCS structure (see supplementary Section I). We also added a schematic of the gate to Fig. 4a inset in the main manuscript. (2) 0-voltage in Figure 4a means $V_g = 0$ V on the gate, not the bias voltage. The x-axis for Fig. 4a is gate voltage.

I respectfully point out to the authors that there is no need to take the comments personal or reply aggressively, my comments were not intended as an aggression. I consider myself a person that is very familiar with terahertz devices, having worked in the field for almost two decades. With all

due respect a clear explanation of how you call your electrodes, perhaps with a clear diagram is something that I consider a reasonable request. Furthermore, in the terahertz community the typical nomenclature for a photoconductive switch is cathode and anode, which are typically the only two electrodes present, since it is a switch, not a mosfet with a gate and a channel. Even if you opt to keep using the FET nomenclature for the electrodes, a clear explanation will make it easy to understand for everyone what is meant. Perhaps names in Fig 1 would get rid of all ambiguities.

First of all, we want to be clear that our previous response is neither personal nor aggressive, and we are sorry if the referee somehow was offended. We absolutely respect the time and energy a referee put in for evaluating the work—we also served as referees ourselves and understand the voluntary commitment. Second, for the final part of the study (Fig 4), because we added the third electrode, gate electrode, it is no longer a typical two-terminal photoconductive switch, and therefore we use gate voltage from the FET nomenclature. For someone who is not familiar with this body of work, it can be confusing, and that is why we point out in our previous response, as an explanation rather than trying to offend the referee. Finally, Fig. 1 has no electrical gate, and we added labels “Probe” (the optical gate) and “Pump” in the second revision to avoid ambiguity; in Fig. 4a we already labelled the bias voltage “U” and the gate voltage “Vg”. Here we use “Probe” for the optical gate beam to avoid confusion with the electrical gate.

7. Between lines 167 and 169, a description of what is seen in Fig 4d is presented, but this does not match what is seen in the figure. For instance the authors discuss voltages between -3V and +7V, but in the figure only positive voltages are shown. Furthermore, the minimum of the "M"-curve appears at 3V for one of the two curves, but not for the other. All in all this part was also confusing, and there are inconsistencies between the text and the plot.

We appreciate the referee's comment and have revised both the text and the figure to clarify the observed trend.

“Fig. 4d shows the field's amplitude captured by the X and Y channels of a lock-in amplifier, which reflects both the absolute value and the phase of the signal. The amplitude's absolute values peak

at around -3 V and 7 V (orange dotted lines), but dips around 3 V (green dotted line), leading to a pair of M-shaped dependence (the red M positioned upward, and the blue M mirrored by the horizontal axis due to the phase of Y channel). ”

Thanks

8. The paragraph between lines 183 and 190, also make claims comparing the results in figure 2 and 3, however, for the reasons already mentioned, about the detector being strongly affected by the carrier lifetime, it is not possible to make fair comparisons.

As we explain in previous response, the detectors in two devices are implanted with the same dose while the PCS channel is doped differently. Hence the comparison is fair. We hope the revised description is clear to our readers.

Thanks

9. Other specific comments:

a. line 12: "To obtain picosecond ultrafast pulses, semiconductors are typically heavily damaged to reduce the carrier lifetime". Please reconsider

Unfortunately, the reviewer’s comment here is incomplete, probably by mistake, so we are not quite sure how to address it. In response, to this comment, we revised the text from ‘heavily damaged’ to ‘with high defect density’.

I believe “heavily damaged” is not a defined term, I understand that you mean ion-implantation, but “heavy damage” could be achieved by dancing on the sample for 20 minutes, or by throwing it from the third floor of a building. Yet, even if we refer to damage by ion implantation the “heavy”

part is completely subjective, is a dose of 1 or 10 or 10^{20} ions/cm² heavy or where do you draw the line?

In our previous revision, we have changed the term “heavily damaged” to “with high defect density”.

b. line 28: "channel", please consider using the more common term "gap"

Thanks for the comment. We feel that using ‘gap’ and ‘channel’ has different benefits throughout the paper. The word ‘gap’ is indeed widely used by convention in many PCS literatures, while our choice of channel better describes our device following the convention of transistors. To avoid confusion, we added the following words to the manuscript:

“(or a gap, as it is often referred to in the literature)”

Ok

c. line 30: idem

Idem means “the same as the previous comment”

d. lines 30-32: please reconsider the sentence starting "The transient current ..."

This is not a clear comment/critique, so we are not quite sure how to address it.

“The transient current in *a* PCS couples with a*n* antenna to emit the THz *electromagnetic radiation* to the far field”

e. line 33: the authors talk about "low quantum efficiency", yet this is not a term that is normally used at least in the THz community, it would be easy to find efficiencies that are not that low since THz photons contain about 3 to 4 orders of magnitude less energy than IR-VIS photons, one could have "quantum efficiencies" close to 1 and still convert 0.1% to 0.01% of the laser energy into terahertz energy. Perhaps "low conversion efficiency" is more appropriate, although in general this is also not a parameter that is typically reported, as the bias voltage also plays a fundamental role in the conversion, so it is not a property of the device as such.

We agree with the referee that "conversion efficiency" is a more relevant term, and have modified the text accordingly.

Thanks

f. line 35: "carrier recombination", I believe the authors refer to carrier trapping, since defects act as traps, but this is a different process to the recombination.

We changed 'carrier recombination' to 'carrier lifetime'.

Thanks

g. line 36-37: the carrier mobility drops, but not by "a few orders of magnitude" please reconsider

We changed it to 'drops substantially'.

Thanks

h. line 42: "reducing the overall device active area" This is actually not entirely true, the patterned antennas actually allow larger laser beams, which results in a larger emitter area, in the case of plasmonic contacts, they act a bit like "light funnels" so in the end the effective area is not necessarily reduced.

We removed the words to avoid any controversy.

Thanks

i. line 73. Although "pump-probe" has been wrongly used by many authors, in general the THz community understands as a pump-probe setup the one where a sample (not the emitter or detector) is optically pumped by a third laser pulse, and probed by the THz pulse. The more accepted, and less ambiguous, term used now is THz-generation (beam) and gate (detector beam).

We understand there are different terminology regarding this within the literature. We do believe our terminology used in this work is fully within the conventions of much of the literature, and is completely unambiguous in the context of our text. In order to remove any possible ambiguity, we have added "or gate" to the text (line 95) to indicate that the probe pulse indeed gates the signal.

Thanks, perhaps names in Fig 1 would get rid of all ambiguities

We added new labels "Probe" and "Pump" in Fig. 1b to avoid ambiguities in the second revision. We want to avoid using "Gate", since it may cause confusion with the gate electrode we used in Fig. 4.

j. The dimensions of the device are not mentioned in the manuscript, please include them

Done

Thanks

k. The repetition rate of the laser and power of each beam are also not mentioned in the device fabrication and characterization section

We've added the repetition rate (76 MHz) in the revised manuscript. The pump and probe power are described in the captions of the relevant figures.

l. Your pulses look indeed Gaussian, but the typical shape of THz waveform pulses is bipolar, or even more complicated, the pulses should integrate zero over time, this is for charge conservation reasons. Perhaps your current signals require further processing, I believe this is related to the carrier lifetime not being short enough in the detector.

The referee is correct in his/her observation that the waveform is not bipolar. In fact, it is not required that the pulse be bipolar with an obvious zero time integral within the experimental scan range. It is a common misconception that this is required, but in fact this is only required in the FAR FIELD of a source. (Of course, the integral must be rigorously zero in the far field, but it can indeed be nonzero near the emitter). This subject has been dealt with in great detail in a classic paper by Phil Bucksbaum and his student D. You back in the 1990's; the reference is:

D. You and P. H. Bucksbaum, "Propagation of half-cycle far infrared pulses," J. Opt. Soc. Am. B 14, 1651-1655 (1997).

That paper shows that THz pulses can appear as quasi-unipolar for quite surprising distances, and this is the case with our experimental pulses. The detection is not fully in the far field, as the signal is confined to a transmission line near the source.

Now, it is true that there must be a negative component that would eventually integrate to zero. However, the amplitude of this component can be so small that it is completely within the noise limit of the data, and correspondingly the negative "tail" can be as long as nanoseconds (corresponding essentially to the re-charging of the semiconductor gap).

We have discussed the issue and added this reference in the supplementary, so that any reader may find this, but a long discussion of the pulse shape would not be necessary, as the above reference is sufficient.

Thanks, I must point out that this does not match my experience.

Indeed free-space radiation generally satisfies the zero-time-integral constraint. We also point out that the discussion from comment #2 is also relevant: when the contacts are ohmic a transmission-line geometry will provide a step function voltage in the case of a long-lifetime semiconductor, or a quasi-unipolar pulse in the case of a short-lifetime switch material. We fully agree free-space pulse will integrate to zero, but this need not be the case on a transmission line.

m. line 128: there si an error (I think a reference)

Corrected.

Thanks

n. line 179: idem.

Corrected.

Thanks

Reviewer #2 (Remarks to the Author):

I have read the responses to my previous comments for the manuscript "Strongly Enhanced THz Emission Enabled by a Graphene Hot-Carrier Fast Lane".

A good number of my concerns have been addressed in the first round of review, and some others in the second round. Yet, I still have some comments that need addressing. I only copy here the third response to the points that will need further clarification or modifications in the manuscript using the original numbering of my comments. The missing numbers have already been addressed in the first or second round of reviews.

1. (FIRST PART) I would like to thank the authors for their response. However, I am not convinced. The actual change of the waveguide in the near field could be the reason for better guiding and therefore more signal in the receiver, without conclusive proof that there is more power being generated.

1. (SECOND PART) I still believe that this limitation is relevant should be clearly stated in the manuscript, acknowledging that the total power generated by a graphene-coated device at ... lets say 9 volts ... is less than you would obtain from a non-graphene-coated device at 15 or 20 volts. Having a highly conductive layer, such as graphene will not allow large bias voltages, and I do not really see any way around it.

2. The authors are partly right in the sense that ohmic contacts do have an effect on the emission, but this does not prevent the formation of the dipole in the semiconductor bulk, the bipolarity of the terahertz pulse caused by the formation of the dipole has been documented in both types of contacts on long-carrier-lifetime substrates (J. Appl. Phys. 110, 023111 (2011))

As for the efficient emitters with interdigitated contacts, SI-GaAs is equally or even more efficient than LT-GaAs as emitter even using simple dipole/pow-tie/etc geometries. It is not true that short-carrier-lifetime is needed for the generation side, it is only needed in the detection side.

4. I would like to respectfully insist. From what is said in the manuscript (and in this responses to my comment number 4) I have no idea of how you estimated the lifetime "transient measurements" or "time-delay measurements" are not an established techniques. Which technique? Under what conditions? Etc. Please comment and include it in the manuscript.

6. Thanks!

9a. "with high defect density" is almost equally ambiguous. What were the doses you used? At what energies? If you have this information, the defect density can be calculated by using the free software <http://www.srim.org/>

9i. Thanks!

Point-by-point Responses to Reviewers' Comments

Reviewers' comments are in **black**; our new responses are in **blue**. Our previous response is indicated in **red** to avoid confusion.

Reviewer #2 (Remarks to the Author):

I have read the responses to my previous comments for the manuscript "Strongly Enhanced THz Emission Enabled by a Graphene Hot-Carrier Fast Lane".

A good number of my concerns have been addressed in the first round of review, and some others in the second round. Yet, I still have some comments that need addressing. I only copy here the third response to the points that will need further clarification or modifications in the manuscript using the original numbering of my comments. The missing numbers have already been addressed in the first or second round of reviews.

1. (FIRST PART) I would like to thank the authors for their response. However, I am not convinced. The actual change of the waveguide in the near field could be the reason for better guiding and therefore more signal in the receiver, without conclusive proof that there is more power being generated.

We respectfully disagree with the referee. We do agree, however, further experimental work will be needed to characterize the far-field emission, and hopefully the publication of this work will inspire similar hot-carrier fast lane designs for THz generation.

In addition, we do have evidence that suggest the enhancement is not simply due to the changes in waveguide. As shown in Fig. 4d, the gate dependence of the detected amplitude is not simply proportional to the channel conductivity change (Fig. 4a).

1. (SECOND PART) I still believe that this limitation is relevant should be clearly stated in the manuscript, acknowledging that the total power generated by a graphene-coated device at ... lets

say 9 volts ... is less than you would obtain from a non-graphene-coated device at 15 or 20 volts. Having a highly conductive layer, such as graphene will not allow large bias voltages, and I do not really see any way around it.

We respectfully disagree with the referee. Comparing two devices under different biasing voltages is not meaningful. In addition, graphene device can withstand high voltages once it is properly passivated, which is well documented in literatures. We add the following discussion in the manuscript:

Previous studies indicated that graphene has a high breaking field of 30-70 kV/cm with optimized sample preparation (IEEE Electron Device Letters, 2011, 32(4): 557-559. and Nano Research, 2016, 9(12): 3663-3670.). This corresponds to a 45-105 V bias on our 15- μm -long channel. We believe that the graphene's high-voltage tolerance does not set a fundamental limitation of the proposed potential applications.

2. The authors are partly right in the sense that ohmic contacts do have an effect on the emission, but this does not prevent the formation of the dipole in the semiconductor bulk, the bipolarity of the terahertz pulse caused by the formation of the dipole has been documented in both types of contacts on long-carrier-lifetime substrates (J. Appl. Phys. 110, 023111 (2011))

In our response in the previous round of review, we emphasized that there is an important difference between free space emission and THz generation in a waveguide. We fully agree with the reviewer that the free space emission is bipolar even with ohmic contacts due to field screening (buildup of a dipole in the switch region). This is well established, including in the paper that the reviewer cites. However, that paper concerns free space radiation. It does not discuss the THz voltage waveform on a transmission line. As we discussed (and cited) in the last round, the presence of ohmic contacts on a coplanar waveguide does not result in a bipolar waveform with zero time-integral, as one has in the far field in free space. The new comment by the reviewer addresses only the free space case, and does not address our actual experimental geometry, and it is very important to keep the distinction in mind!

As for the efficient emitters with interdigitated contacts, SI-GaAs is equally or even more efficient than LT-GaAs as emitter even using simple dipole/pow-tie/etc geometries. It is not true that short-carrier-lifetime is needed for the generation side, it is only needed in the detection side.

We did specifically add discussion in the last draft, following the referee's previous comment, that SI-GaAs can be used as efficient THz generator even with a long carrier lifetime. We are copying that discussion below. And again, it is important to note the distinction between THz generation on waveguides and in free space.

“While substrates with longer carrier lifetime (such as semi-insulating GaAs) can be used for efficient broadband THz emitter though clever engineering (such as the use of interdigitated antenna)⁷, short carrier lifetimes (picosecond) are in general preferred for efficient THz generation when the PCS is coupled with transmission lines/waveguides for THz modulation, studies of device transient phenomena, or almost any other application employing confined fields.”

4. I would like to respectfully insist. From what is said in the manuscript (and in this responses to my comment number 4) I have no idea of how you estimated the lifetime “transient measurements” or “time-delay measurements” are not an established techniques. Which technique? Under what conditions? Etc. Please comment and include it in the manuscript.

We thank the referee and have added the following discussion to further clarify the measurement technique we used:

The lifetime of transient carriers in silicon Auston switch is characterized with the same on-chip pump-probe measurement with identical layout to our main measurements, with the only difference that the emitter does not have a graphene layer on top. Supplementary Fig. 2c shows the pulse generated by the simple silicon emitter. The observed pulse has a FWHM of 0.9 ps, corresponding to a carrier lifetime of 0.65 ps by deconvolving the emitter and detector transients assuming exponential decays. The value matches well with the value (0.6 ps) observed in samples prepared with identical conditions in previous literatures (ref [29][30]).

6. Thanks!

9a. “with high defect density” is almost equally ambiguous. What were the doses you used? At what energies? If you have this information, the defect density can be calculated by using the free software <http://www.srim.org/>

We thank the referee for the comment. We feel it may be too extensive to go into details in the abstract section. Instead, we made the following changes to the device fabrication section:

“The silicon was first damaged by ion implantation with O⁺ ions at 100 keV with a flux of 10¹⁵ cm⁻², resulting in a defect density of 0.03 average displacement-per-atom, as verified by a previous work with identical sample preparation conditions*.”

* Lui K P H, Hegmann F A. Fluence-and temperature-dependent studies of carrier dynamics in radiation-damaged silicon-on-sapphire and amorphous silicon[J]. Journal of applied physics, 2003, 93(11): 9012-9018.

9i. Thanks!